# Comprehensive characterization of humic-like substances in smoke PM$_{2.5}$ emitted from the combustion of biomass materials and fossil fuels

Xingjun Fan[1,2], Siye Wei[1,3], Mengbo Zhu[1,3], Jianzhong Song[1], Ping'an Peng[1]

[1]State Key Laboratory of Organic Geochemistry, Guangzhou Institute of Geochemistry, Chinese Academy of Sciences, Guangzhou 510640, P. R. China

[2]College of Resource and Environment, Anhui Science and Technology University, Anhui 233100, P. R. China

[3]Graduate School of Chinese Academy of Sciences, Beijing 100049, P. R. China

*Correspondence to:* Jianzhong Song (songjzh@gig.ac.cn)

**Abstract.** Humic-like substances (HULIS) in smoke PM$_{2.5}$ emitted from the combustion of biomass materials (rice straw, corn straw, and pine branch) and fossil fuels (lignite coal and diesel fuel) were comprehensively studied in this work. The HULIS fractions were first isolated with a one-step solid phase extraction method and were then investigated with a series of analytical techniques: elemental analysis, total organic carbon analysis, UV-vis spectroscopy, excitation–emission matrix (EEM) fluorescence spectroscopy, fourier transform infrared spectroscopy, and $^1$H-nuclear magnetic resonance spectroscopy. The results show that HULIS account for 11.2–23.4% and 5.3% of PM$_{2.5}$ emitted from biomass burning (BB) and coal combustion, respectively. In addition, contributions of HULIS-C to total carbon and water soluble carbon in smoke PM$_{2.5}$ emitted from BB are 8.0–21.7% and 56.9−66.1%, respectively. The corresponding contributions in smoke PM$_{2.5}$ from coal combustion are 5.2% and 45.5%, respectively. These results suggest that BB and coal combustion are both important sources of HULIS in atmospheric aerosols. However, HULIS in diesel soot only accounted for ~0.8% of the soot particles, suggesting that vehicular exhaust may not be a significant primary source of HULIS. Primary

HULIS and atmospheric HULIS display many similar chemical characteristics, as indicated by the instrumental analytical characterization, while some distinct features were also apparent. A high spectral absorbance in the UV-vis spectra, a distinct band at λex/λem ≈ 280/350 nm in EEM spectra, lower H/C and O/C molar ratios, and a high content of [Ar–H] were observed for primary HULIS. These results suggest that primary HULIS contains more aromatic structures, and a lower content of aliphatic and oxygen-containing groups than atmospheric HULIS. Among the four primary sources of HULIS, HULIS from BB had the highest O/C molar ratios (0.43–0.54) and [H–C–O] content (10%–19%), indicating that HULIS from this source mainly consisted of carbohydrate and phenolic like structures. HULIS from coal combustion had a lower O/C molar ratio (0.27), and a higher content of [Ar–H] (31%), suggesting that aromatic compounds were extremely abundant in HULIS from this source. Moreover, the absorption Ångström exponents of primary HULIS from BB and coal combustion were 6.7–8.2 and 13.6, respectively. The mass absorption efficiencies of primary HULIS from BB and coal combustion at 365 nm (MAE$_{365}$) were 0.97–2.09 and 0.63 m$^2$/gC, respectively. Noticeable higher MAE$_{365}$ for primary HULIS from BB than coal combustion indicates the former one has stronger contribution to the light absorbing properties of aerosols in the atmospheric environment.

## 1. Introduction

In recent decades, many studies have investigated the water soluble unresolved polyacidic compounds in atmospheric aerosols, rainwater, and fog/cloud samples (Zheng et al., 2013 and references therein). Due to their similarities to naturally occurring humic substances in terrestrial and aqueous environments, with regard to their complex physical and chemical properties, as revealed by techniques such as UV-vis spectroscopy, fluorescence spectroscopy, Fourier transform infrared (FTIR) spectroscopy, and nuclear magnetic resonance (NMR) spectroscopy, they are operationally defined as humic-like substances (HULIS) (Graber and Rudich, 2006; Zheng et al., 2013). HULIS are present ubiquitously in fine particles from urban, rural, marine, and biomass burning (BB) sources (Decesari et

al., 2007; Salma et al., 2007; Lin et al., 2010b; Fan et al., 2012; Song et al., 2012; Zheng et al., 2013). They are believed to play important roles in several atmospheric processes, including light absorption, radiative forcing (Hoffer et al., 2006; Dinar et al., 2008), hygroscopicity, and cloud droplet formation (Dinar et al., 2007; Salma et al., 2008). Moreover, they are also reported to be harmful to human health (Lin and Yu, 2011).

Many field studies have suggested that HULIS are abundant in organic aerosols. They constitute a significant portion of the organic matter (OM) in atmospheric aerosols (up to about 30%) collected in urban and rural environments, and in aerosols produced by BB (Mayol-Bracero et al. 2002; Krivacsy et al. 2008; Lin et al. 2009, 2010). Their carbon (C) mass accounts for 9%−72% of the C content of water soluble organic matter (WSOM) in atmospheric aerosols (Feczko et al., 2007; Krivacsy et al., 2008; Lin et al., 2010b; Fan et al., 2012; Song et al., 2012). These atmospheric HULIS materials are found ubiquitously in various environments, and are derived from various sources. Their possible sources include: biomass burning (BB) (Feczko et al., 2007; Baduel et al., 2010; Lin et al., 2010a), vehicular emissions (El Haddad et al., 2009), marine emissions (Krivacsy et al., 2008), the oxidation of soot (Decesari et al., 2002; Li et al., 2013; Li et al., 2015), and secondary processes via the transformation of gas and condensed-phase species by chemical reactions (Salma et al., 2007; Baduel et al., 2010; Salma et al., 2013).

Among the various sources listed above, BB is generally considered to be a significant source of atmospheric HULIS (Schmidl et al., 2008a; Schmidl et al., 2008b; Goncalves et al., 2010; Lin et al., 2010a; Lin et al., 2010b). HULIS fractions have been found in smoke particles emitted from the combustion of wood and leaves, the carbon content of HULIS (HULIS-C) make up 0.6–21.2% of the total mass of particles (Schmidl et al., 2008a; Schmidl et al., 2008b; Goncalves et al., 2010). HULIS have also been found to be abundant in fresh burning emissions from rice straw and sugar cane leaves (Lin et al., 2010a; Lin et al., 2010b). HULIS accounted for 7.6–12.4% of the particle mass, and HULIS-C contributed approximate 14.3–14.7% and 30–33% of the organic carbon (OC) and the water

soluble carbon (WSOC), respectively. Unfortunately, these studies have only focused on the amount of HULIS emitted from BB, with their chemical properties and structures remaining unresolved.

One recent study, the chemical and light absorption properties of HULIS in $PM_{2.5}$ from burning of three different types of biomass burning fuels (rice straw, pine needles, and sesame branch) in a laboratory combustion chamber were investigated by Park and Yu (2016). According to this study, primary HULIS from BB accounted for 15.3–29.5% of $PM_{2.5}$ emissions, and HULIS-C contributed 15–29% of OC and 36–63% of WSOC, respectively. Although the study brought a better understanding on light absorption properties of primary WSOC from BB, but the observations of the optical and structural features of primary HULIS are limited (Park and Yu, 2016). On the other hand, as important energy resources, fossil fuels (such as coal, diesel fuel, etc.) are consumed significantly around the world, and are known to be important sources of black carbon in ambient aerosols (Cao et al., 2006). However, the contents and chemical properties of primary HULIS from fossil fuels combustion are still unknown.

In this study, smoke $PM_{2.5}$ emitted from the combustion of biomass materials (including rice straw, corn straw, and pine branch) and fossil fuels (including lignite coal and diesel fuel) were collected in a laboratory chamber. The HULIS fractions were isolated from smoke $PM_{2.5}$ by a solid phase extraction (SPE) method, and the chemical properties and structures were comprehensively investigated using total organic carbon (TOC) analysis, elemental analysis, UV-vis spectroscopy, excitation-emission matrix (EEM) fluorescence spectroscopy, FTIR spectroscopy, and [1]H-NMR spectroscopy. To further understand the contributions of primary HULIS to atmospheric HULIS, the HULIS fractions isolated from ambient $PM_{2.5}$ collected in Guangzhou, China, were simultaneously studied and compared with the above primary HULIS. Moreover, the results obtained were also compared with those reported in the literature for HULIS in various atmospheric environments. The information obtained will enable a better understanding of the chemical nature, as well as the environmental, health, and climate effects of primary HULIS from direct combustion emissions, and their contribution to atmospheric HULIS.

## 2 Experimental

### 2.1 Sampling

In this study, five types of smoke $PM_{2.5}$ samples were collected to investigate the primary HULIS emitted from the combustion of biomass materials and fossil fuels. These were biomass smoke $PM_{2.5}$ samples emitted from the combustion of rice straw, corn straw and pine branch, and coal smoke $PM_{2.5}$ and diesel soot. Rice straw and corn straw were chosen primarily because rice and corn are the dominant crops in China. The combustion of these crop straws is reported to have an important influence on the atmospheric aerosol in China (Streets et al., 2003; Zheng et al., 2006). These crop residues are usually burned in open locations by farmers during and after the harvest season, and are also used as cooking fuels in rural areas throughout the year. In addition, pine branches are also important biomass cooking fuel in rural areas of China and therefore may make a significant contribution to the atmospheric aerosol in some regions. Therefore, samples of the smoke emitted from the combustion of these three biomass materials were used to study the BB derived HULIS. Samples of the smoke emitted from the combustion of coal and diesel fuel were also studied in this work. Coal was chosen because it has been reported that more than 68% of black carbon (BC) emissions in China are related to the use of coal (Cao et al., 2006). The combustion of coal is an important source of atmospheric aerosols in China. In addition, the soot particles derived from the combustion of diesel fuel was also studied because vehicular emissions have been suggested to be a possible source of atmospheric HULIS (El Haddad et al., 2009).

The biomass materials including rice straw, corn straw, pine branches were collected from rural area of Guangdong province, and the coal (the vitrinite reflectance ($R_o$) is 0.77%) were obtained from Ping Ding Shan, China. The detail information on this type of coal could be found in Huang et al. (2013). The ultimate properties of the three biomass materials and coal are shown in Table 1. On an air-dry basis, moisture content measured for the rice straw, corn straw, pine branch and coal was 5.8 $\pm$ 0.5, 7.4 $\pm$ 0.8, 7.6 $\pm$ 0.7, and 1.6 $\pm$ 0.2 %, respectively. Carbon (C), hydrogen (H), and oxygen (O)

contents were found to range from 36.0 to 72.6%, 4.0 to 7.2%, and 8.2 to 45.0% for combustion materials, respectively. In comparison with biomass materials, coal substantially composed of higher C content (72.6%) and lower O content (8.2%). There were no significant differences among biomass materials in terms of elemental compositions.

In this study, samples of the smoke $PM_{2.5}$ emitted from the combustion of rice straw, corn straw, pine branch, and lignite coal were collected in a laboratory resuspension chamber. This sampling system included a combustion stove and two $PM_{2.5}$ samplers (Tianhong Intelligent Instrument Plant, Wuhan, China). The instrument is described in detail in Duan et al. (2012). The combustion experiments of biomass fuels (rice straw, corn straw and pine branch) were carried out under open air without any controlled conditions to simulate open burning in the field. Smoke $PM_{2.5}$ samples were collected on Whatman quartz filters (Ø 90 mm) by two samplers in the chamber. For each biomass combustion experiment, biomass materials were first cut into pieces, and then were ignited and burned out, and one set of two smoke $PM_{2.5}$ filters were collected during whole burning process (5~15 min). Totally, five sets of filters samples were collected for each biomass fuel. The coal combustion was carried out according to the method introduced by Huang et al. (2013). The combustion stove was put into the chamber when the combustion condition was stabilized, and then one set of smoke $PM_{2.5}$ sample was collected for approximately 10 min, and a total of five sets of filter samples were obtained. For soot, a reference sample (SRM 2975) from the combustion of diesel fuel was purchased from the U.S. National Institute of Standards and Technology (Gaithersburg, MD, USA). It was emitted from a heavy-duty diesel engine and represents diesel fuel combustion from a vehicular exhaust.

In addition to the smoke $PM_{2.5}$ samples emitted directly from the combustion process, ambient $PM_{2.5}$ samples were also collected during December 7 to 11, 2015 in Wushan, Guangzhou, China. Each sample was collected for approximately 24 h, and a total of 5 filters were obtained. Detailed information regarding the sampling sites was provided in our previous studies (Fan et al., 2012; Song and Peng, 2009). These $PM_{2.5}$ samples were collected on Whatman quartz fiber filters (20.3 × 25.4 cm) using a

high-volume air sampler at flow rates of 1.05 m$^3$/min (Tianhong Intelligent Instrument Plant, WuHan, China). All filters had been pre-baked at 450°C for 4 h to remove all organic contaminants.

## 2.2 Isolation of HULIS

In this study, to obtain accurate results for TC, HULIS, WSOM, and other parameters, five filters for each type of sample were selected for analysis. The isolation of HULIS was performed by a one-step SPE procedure, which was applied by many studies (Kiss et al., 2002; Lin et al., 2010a,b; Fan et al., 2012; Park and Yu, 2016). Briefly, 3 cm$^2$ filter samples were ultrasonically extracted with 40 mL of 18.2 MΩ Milli-Q water, and then the extract solutions were filtered with polytetrafluoroethylene (PTFE) membranes (pore size: 0.22 μm) to remove solid impurities and filter debris. The pH value of the filtrate was adjusted to 2 with HCl, and then 20 mL was introduced into a pre-conditioned SPE cartridge (Oasis HLB, 500mg, Waters, Milford, MA, USA). The exposed column was rinsed with water to remove inorganics and dried in a freeze-drier. Finally, the retained organics were eluted with methanol and the eluate was evaporated to dryness under a gentle nitrogen stream. According to the requirements of TOC, UV-vis spectroscopy, and EEM fluorescence spectroscopy analysis, the resulting HULIS samples were re-dissolved in 20 mL of Milli-Q water. Moreover, more area of the corresponding filters was used to obtain enough dried HULIS for the analysis of the elemental composition, as well as FTIR and $^1$H NMR spectrometry.

It is noted that the eluates here represent the hydrophobic portion of WSOM and were named as water soluble HULIS. According to the literatures (Graber and Rudich, 2006; Zheng et al., 2013), these water soluble hydrophobic WSOM can be isolated with different SPE methods. In spite of some differences that were observed among of them, these hydrophobic WSOM isolated with different sorbents are very similar in chemical compositions and properties according to our previous studies (Fan et al., 2012, 2013). Therefore, for better comparison with other studies, the hydrophobic WSOM isolated by SPE methods (i.e. HLB, C-18, DEAE, XAD-8) and other protocols (i.e. ELSD) are all

termed as HULIS in this paper.

## 2.3 Analysis

### 2.3.1 Total carbon (TC) and TOC analysis

The TC content of smoke $PM_{2.5}$ was measured directly on 2 $cm^2$ punches of the particle filters using an elemental analyzer (Elementar Vario EL CUBE, Hanau, Germany), following a standard high-temperature combustion procedure. A replicate analysis was also conducted for accuracy. The TOC in HULIS and WSOM were measured using a high-temperature catalytic oxidation instrument (Shimadzu – TOC – VCPH analyzer, Shimadzu, Kyoto, Japan) following the non-purgeable organic carbon protocol. The concentrations of all chemically measured species were corrected for their respective blank concentration.

### 2.3.2 Elemental composition

Elemental composition (C, H, N) of the isolated HULIS was measured with an elemental analyzer following a standard high temperature combustion procedure. A portion of the HULIS (re-dissolved in methanol) was transferred into a pre-cleaned tin capsule of known weight. Then, the sample was dried under vacuum. The mass of the dried organic matter (OM) in the tin capsule was determined using a micro balance, with a resolution of 0.01 μg (Sartorius, Göttingen, Germany), and then the tin capsule was placed into the elemental analyzer. In the instrument, the C, H, and N content of the OM were determined by catalytic burning in oxygen at 1020$^o$C, followed by chromatographic separation of the oxidation products and thermoconductivity detection. The elemental analyzer was calibrated with an acetanilide standard. Based on the analyses of triplicates for each sample, the calculated relative standard deviation was less than 3%. The O content was calculated as the rest of the mass, by assuming that the concentrations of other possible elements (e.g., sulfur, phosphorus) were negligible: O% = 100 − (C+H+N)%.

### 2.3.3 UV–visible spectroscopy

About 3 mL of HULIS and WSOM solution were placed in a 1 cm quartz cuvette and scanned from 200 to 700 nm using a UV-vis spectrophotometer (Lambda 850, Perkin Elmer, Waltham, MA, USA). Milli-Q water was used as a blank reference and to obtain the baseline. The absorption at 250 nm (UV250), absorption index, and light absorption properties were determined to characterize the optical properties of HULIS. They are described as following:

(1) The absorption at 250 nm (UV250)

As demonstrated by many studies, the high absorbing UV chromophoric compounds (strong absorbing at 250 nm) are major components in WSOM, which usually tended to be enriched in the SPE isolated HULIS fractions (Baduel et al., 2009; Fan et al., 2012, 2013, 2016; Song et al., 2012; Duarte et al., 2015; Lopes et al., 2015; Fan et al., 2016). Therefore, the ratio between the UV250 of HULIS and original WSOM has been widely used to evaluate the relative contribution of HULIS to WSOM in terms of chromophoric compounds content. It should be noted that HULIS solution must keep the same volume to original WSOM solution for the UV250 determination.

(2) Absorption index

The specific UV-vis absorbance at 254 nm, which is normalized by DOC of solution, and absorptivity ratios between 250 and 365 nm ($E_{250}/E_{365}$) have been successfully applied to characterize the chemical properties of HULIS and WSOC. They are also determined in this work, and the details can be found in our previous studies (Fan et al., 2012,2016).

(3) Light absorption properties

HULIS have been proved to have strong wavelength dependence with absorption increasing sharply from the visible to UV ranges (Hecobian et al., 2010; Park and Yu, 2016). In this study, the absorption Ångström exponent (AAE) and the mass absorption efficiencies at 365 nm ($MAE_{365}$) were calculated based on UV-vis spectroscopy analysis to investigate the light absorption properties of

HULIS samples.

AAE is a measure of the spectral dependence of light absorption from chromophores in HULIS. In this study, the AAE were calculated based on the linear regression fit of logarithms of $A_\lambda$ and wavelength between 330 and 400 nm, according to the following power law equation:

$$A_\lambda = K\lambda^{-AAE}$$

where, $A_\lambda$ is the absorbance derived from the spectrophotometer at a given wavelength $\lambda$, K is a constant.

MAE ($m^2/g$) is a key parameter that describes the light absorbing ability of different chromophores. In this study, the MAE at 365 nm ($MAE_{365}$) was used to characterize the light-absorbing ability for HULIS, and was calculated using the following equation (2):

$$MAE = \frac{A_\lambda}{C \cdot L} \times ln(10)$$

where, $C$ is the DOC content of HULIS in solution (ugC/mL), $L$ is the optical path length (0.01 m).

### 2.3.4 EEM fluorescence spectroscopy

The fluorescence spectra of each HULIS sample were recorded on a spectrophotometer (F-2700, Hitachi, Tokyo, Japan), using a 1 cm path-length quartz cuvette. Excitation and emission wavelength ranges were set from 210 to 400 nm and 230 to 510 nm, respectively, and their scanning intervals were all set at 5 nm. The excitation and emission slit widths were fixed at 5 nm and the scan speed was set at 1500 nm/min. The peaks due to water Raman scatter were eliminated from all sample EEMs by subtracting the Milli-Q water blank EEMs.

### 2.3.5 FT-IR Spectrometry

The FTIR spectra (4000–400 $cm^{-1}$) of HULIS were recorded at room temperature using an FTIR spectrometer (Vertex-70, Bruker, Mannheim, Germany). About 1 mg of HULIS (re-dissolved in

methanol) was first mixed with 60 mg of KBr, and then dried in a freeze dryer. Finally, the above mixture was grated and pressed into pellets for analysis. For each measurement, 64 scans were collected at a resolution of 2 cm$^{-1}$.

### 2.3.6 $^1$H-NMR spectroscopy

About 10 mg of dried HULIS were dissolved in deuterated methanol (MeOH-d$_4$, 1 mL) and transferred to 5 mm NMR tubes. The $^1$H NMR spectra of HULIS were recorded on a 400 MHz NMR spectrometer (Avance III, Bruker). For each sample, 128 scans were collected, resulting in an analysis time of approximately 1 h. The identification of functional groups in the NMR spectra was based on their chemical shift ($\delta$H) relative to that of sodium 3-trimethylsilyl-2,2,3,3-$d_4$-propanoate ($\delta$H= 0 ppm), which was used as an internal standard.

## 3 Results and discussion

### 3.1 The abundance of HULIS in smoke PM$_{2.5}$ and ambient PM$_{2.5}$

The average abundance of the HULIS fractions and their contributions to particle matter (PM), TC, and WSOM in smoke PM$_{2.5}$ emitted from the combustion of biomass materials, fossil fuels, and in ambient PM$_{2.5}$ are shown in Table 2. It can be seen that the mass of the HULIS fractions accounted for 11.2–23.4% of the PM in smoke PM$_{2.5}$ emitted from BB, which is comparable to the results (7.6–29.5%) for BB reported in previous studies (Lin, 2010b, Park and Yu, 2016). It is worth noting that the highest HULIS abundance (23.4 $\pm$ 5.5%) was detected in rice straw smoke PM$_{2.5}$, which is consistent with 29.5% for similar samples observed by Park and Yu (2016). But it is approximately double the 12.4% reported for similar samples by Lin et al. (2010b), which may be ascribed to their different combustion conditions and sampling methods. The abundance of HULIS in rice straw smoke PM$_{2.5}$ (23.4 $\pm$ 5.5%) was also significantly higher than in ambient PM$_{2.5}$ in this study, and in some previous studies (as listed in Table 2). The abundance of HULIS in smoke PM$_{2.5}$ emitted from the combustion of corn straw and

pine branch (11.2 $\pm$ 7.5% and 11.4 $\pm$ 3.8%, respectively) was significantly lower than in smoke PM$_{2.5}$ from rice straw. This was similar to the results for ambient PM$_{2.5}$ in this study and values reported by Lin et al (2010b), although they were higher than the 6.7% reported by Salma et al. (2007) and an annual observation (5.4%) in the same location, reported in our previous study (Fan et al., 2016).

In comparison with the three BB smoke PM$_{2.5}$ samples, the relative contribution of HULIS in coal smoke PM$_{2.5}$ was relatively low (5.3 $\pm$ 0.4%). This was significantly lower than the level in PM$_{2.5}$ in this study and that reported by Lin et al (2010), but was similar to the annual average result (5.4%) for ambient PM$_{2.5}$ in our previous study (Fan et al., 2016). According to data reported by Cao et al. (2006), more than 68% of BC emissions in China are related to the use of coal. Therefore, it is expected that coal combustion is an important primary source of atmospheric HULIS. It is noteworthy that HULIS only accounts for ~0.8% of diesel soot, suggesting that the primary source of atmospheric HULIS from vehicle exhaust may be negligible. Because the abundance of HULIS in diesel soot were so low, it was difficult to characterize this HULIS fraction.

As an important carbonaceous component, the C content of HULIS (HULIS-C) in smoke PM$_{2.5}$ particles emitted from the combustion of rice straw, corn straw, and pine branch accounted for 21.7 $\pm$ 4.0, 14.7 $\pm$ 6.9, and 8.0 $\pm$ 2.9% of the TC, respectively. These results are very consistent with the results reported for BB in previous studies (Schmidl et al., 2008a, b; Goncalves et al., 2010; Lin et al., 2010b; Park and Yu, 2016). The HULIS-C in coal smoke PM$_{2.5}$ was 5.2 $\pm$ 0.3%, which was significantly lower than that (22.6 $\pm$ 3.7%) in ambient PM$_{2.5}$. The contribution of HULIS-C to the TC of diesel soot was about 0.7%, which was similar to the 1.0%−1.3% reported for vehicular exhaust in El Haddad et al. (2009). These very low HULIS-C/TC ratios also suggest that the primary vehicular exhaust source for atmospheric HULIS may be negligible.

It is well known that HULIS is an important component in WSOC. In this study, the contribution of HULIS to WSOC was investigated by the determination of TOC and UV250, respectively. As indicated in Table 2, the HULIS/WSOM ratios in the five types of smoke PM$_{2.5}$ were 45.5%−66.1%

(TOC, %) and 58.3%−79.5% (UV250, %), indicating that HULIS was the major component in WSOM for the samples studied here. These percentages lie within the range of 19%−72% that has been reported for fine aerosols in many earlier studies (as listed in Table 2). Although relatively high HULIS/WSOM ratios were observed for all of the different smoke PM$_{2.5}$, some differences were apparent. The HULIS/WSOM for the three types of biomass smoke PM$_{2.5}$ was 56.9%−66.1 and 68.1%−79.5%, as determined by TOC and UV250, respectively. They are similar to the values obtained for atmospheric HULIS in this study, at 60.7 and 70.7%, respectively. These results are also comparable with those reported for ambient aerosols (Zheng et al., 2013 and references therein). The HULIS/WSOM was relatively low for coal smoke PM$_{2.5}$, at 45.5 and 64.4%, as determined by TOC and UV250, respectively. For the diesel soot, despite the very low HULIS-C/TC, the HULIS/WSOM for diesel soot was relatively high, at 62.3 and 58.3% as determined by TOC and UV250, respectively. It should be noted that the values of HULIS/WSOM determined by UV250 were mostly higher than those obtained by TOC. These differences have been derived from similar measurement methods and have been reported in many previous studies (Baduel et al., 2009; Fan et al., 2012), mainly as the result of the enrichment of highly conjugated π bond compounds in HULIS fractions. A more detailed explanation is given in section 3.3.

### 3.2 Elemental composition

The elemental compositions (C, H, N, and O) of HULIS in smoke PM$_{2.5}$ from the combustion of rice straw, corn straw, pine branch, and coal, and in ambient PM$_{2.5}$ are shown in Table 3. The mean C, H, N, and O contents for the primary HULIS samples were 52.9−66.1%, 5.5−7.1%, 2.0−4.1%, and 23.6−38.3%, respectively, by mass. This indicates the dominance of C and O, which together contributed 89.7%−91.4% of the total mass. It is obvious that the primary HULIS contain substantially higher C and lower O than ambient HULIS in this study. However, these results are comparable with those for atmospheric HULIS in previous studies (Krivacsy et al., 2001; Kiss et al., 2002; Duarte et al., 2007; Salma et al., 2007; Song et al., 2012; Fan et al., 2013; Duarte et al., 2015).

Only a limited amount of information regarding the different HULIS could be obtained from the elemental composition data, but more qualitative information was obtained by an examination of the O/C, H/C, and N/C molar ratios. These atomic ratios are often used to describe characteristic and structural changes of organic macromolecules (Duarte et al., 2007; Salma et al., 2007; Fan et al., 2013). The O/C molar ratios for the primary HULIS ranged from 0.27 to 0.54, indicating the existence of O containing functional groups. However, these ratios were significantly lower than 0.61–0.68 for the standard fulvic acids of the International Humic Substances Society (Duarte et al., 2007), indicating that primary HULIS was less oxidized when compared to the fulvic acids. Among the four types of primary HULIS, the O/C ratios of the three types from BB were in the range of 0.43–0.54, which were lower than 0.65 for ambient HULIS in this study, but were comparable with data (0.30–0.76) for atmospheric HULIS reported in previous studies (Krivacsy et al., 2001; Kiss et al., 2002; Duarte et al., 2007, 2015; Salma et al., 2007; Song et al., 2012; Fan et al., 2013). The O/C ratio of HULIS in coal smoke $PM_{2.5}$ was 0.27, which was significantly lower than for atmospheric HULIS samples. These results suggest that HULIS in the fresh coal smoke $PM_{2.5}$ could be regarded as less oxidized than HULIS in ambient aerosols. The H/C molar ratios of the four primary types of HULIS were in the ranges of 1.15 to 1.43, which were lower than that (1.59) for atmospheric HULIS in this work. However, they fell in the range of observations (1.01–1.53) reported in previous studies (Krivacsy et al., 2001; Kiss et al., 2002; Duarte et al., 2007, 2015; Salma et al., 2007; Song et al., 2012; Fan et al., 2013). The N/C molar ratios of primary HULIS were 0.03–0.06, with both being similar to the results for atmospheric HULIS in this and previous studies (Table 3). In addition, the ratio of OM to organic C (OM/OC) mass ratios of the four types of primary HULIS ranged from 1.51 to 1.89, which were lower than 2.06 for ambient HULIS in this study, but were generally in the range of the data (1.5–2.28) reported for atmospheric HULIS in previous studies (Krivacsy et al., 2001; Kiss et al., 2002; Duarte et al., 2007; Salma et al., 2007; Song et al., 2012; Fan et al., 2013; Duarte et al., 2015).

In summary, the four types of primary HULIS among the smoke $PM_{2.5}$ samples had many

similarities, in terms of its elemental composition, to atmospheric HULIS samples. However, there were also some distinct differences. The HULIS samples in the three types of biomass smoke $PM_{2.5}$ had a relatively lower C content (52.9–57.4%), higher O content (33.0–38.3%), higher O/C molar ratio (0.43–0.54) and higher OM/OC (1.74–1.89) than those in coal smoke $PM_{2.5}$. These results indicated that the HULIS in BB smoke contained a relatively higher content of O-containing components. Among the three HULIS in BB smokes, the O/C molar ratio of corn straw HULIS were higher than those of rice straw and pine branch HULIS, indicating it contain more contents of O-containing compounds. In terms of H/C molar ratio, HULIS from rice straw and coal combustion exhibit lower values than other two primary HULIS.

### 3.3 UV-vis properties

UV-vis absorbance has been widely used to characterize the properties of organic matter in soils, waters, and atmospheric systems. Figure 1 shows the UV-vis spectra of the four primary types of HULIS in smoke $PM_{2.5}$ emitted from BB and coal combustion, and one atmospheric HULIS sample from ambient $PM_{2.5}$. The spectra were all normalized by the C content of HULIS to avoid the effects of different concentrations and were easily compared with each other. As shown in Figure 1, the UV-vis spectra of all HULIS fractions were featureless, while they displayed a generally decreasing absorbance as the wavelength increased. Such spectra are similar to the typical UV-vis spectra of atmospheric HULIS in previous studies (Havers et al., 1998; Varga et al., 2001; Kiss et al., 2002; Duarte and Duarte, 2005; Duarte et al., 2005; Krivacsy et al., 2008; Baduel et al., 2009, 2010; Fan et al., 2012) and in naturally occurring humic substances (Traina et al., 1990; Peuravuori and Pihlaja, 1997; Chen et al., 2002; Domeizel et al., 2004). These results suggest that the primary HULIS in smoke $PM_{2.5}$ has a similar chemical structure to HULIS in atmospheric aerosols.

Although the spectra appeared to be broad and featureless, some differences in the absorption intensity were apparent. The spectra obtained for the four types of primary HULIS in smoke $PM_{2.5}$

exhibited a higher normalized absorbance in shorter wavelength regions, and less absorbance in the longer wavelength regions than atmospheric HULIS. A clear shoulder in the region 250 to 300 nm was observed in the spectra of primary HULIS fractions in smoke $PM_{2.5}$ emitted from BB and coal combustion. This is generally attributed to $\pi$-$\pi^*$ electron transitions in moieties containing C=C and C=O double bonds, and is also a characteristic of fulvic acids (Peuravuori and Pihlaja, 1997; Domeizel et al., 2004). These results suggest that the primary HULIS in smoke $PM_{2.5}$ may contain a higher concentration of polycyclic aromatic and conjugated compounds than ambient HULIS.

Among of the four types of primary HULIS, HULIS in smoke $PM_{2.5}$ emitted from corn straw burning had a higher normalized UV-vis absorbance in the overall spectra than the other primary HULIS fractions. HULIS in coal smoke had a lower normalized absorbance than the other types of primary HULIS, and was even much lower than that for atmospheric HULIS in the region of 350 to 535 nm, with its absorbance mainly focused in region of 235 nm to 385 nm. This may suggest that polycyclic aromatic and/or conjugated compounds are the most important components in this type of HULIS, and was in strong agreement with the results obtained from the [1]H NMR analysis.

The $SUVA_{254}$ and $E_{250}/E_{365}$ have been found to be correlated with aromaticity and molecular weight of natural occurring humic acids (Peuravuori et al., 2001; Fuentes et al., 2006; Chen et al., 2002). They have also been frequently applied to characterize HULIS in atmospheric aerosols (Duarte and Duarte, 2005; Duarte et al., 2005; Krivacsy et al., 2008; Baduel et al., 2009, 2010; Fan et al., 2012). In this study, these parameters were used to perform comparisons between HULIS in smoke $PM_{2.5}$ and in ambient $PM_{2.5}$, with the results shown in Table 4. The $SUVA_{254}$ values of primary smoke HULIS samples ranged from 3.7 to 3.9 L (m mgC)$^{-1}$, which was higher than the value of the atmospheric HULIS fractions obtained in this study and our previous studies (Fan et al., 2012, 2016). These results indicate that the primary HULIS contain more aromatic groups with conjugation of $\pi$-bonds alongside aliphatic structures. Similar characteristics were also found in many previous studies. For example, it has been found that HULIS fraction in colder season presented more aromatic structures than those in

warmer season, of which the BB might be an important contribution of the former one (Baduel et al., 2010; Matos et al., 2015a, b; Paula et al., 2016). Moreover, the $E_{250}/E_{365}$ ratios were also investigated, which were found to be 5.8 $\pm$ 0.5, 4.5 $\pm$ 0.2, 4.4 $\pm$ 0.3, and 14.7 $\pm$ 0.7 for primary HULIS emitted from the combustion of rice straw, corn straw, pine branch, and coal, respectively. The $E_{250}/E_{365}$ ratio generally exhibits a negatively relationship with the aromaticity or molecular weight of humic-like substances (Duarte and Duarte, 2005; Fan et al., 2012). The $E_{250}/E_{365}$ ratios of primary HULIS in smoke $PM_{2.5}$ from BB were in the range of 4.4–5.8, which is comparable to that (2.9–8.9) of atmospheric HULIS in here and previous studies (As listed in Table 4).

There were also some distinct features among the different types of primary HULIS in terms of their UV-vis properties. No significant differences in $SUVA_{254}$ were identified among the primary HULIS from the combustion of rice straw, corn straw, pine branch, and coal. However, the $E_{250}/E_{365}$ ratios of the four types of primary HULIS ranged from 4.4 to 14.7, with the highest values for the HULIS from coal combustion. It was noteworthy that the $E_{250}/E_{365}$ ratio of HULIS in coal soot was 14.7, which was much higher than for the HULIS in biomass smoke $PM_{2.5}$ and in atmospheric $PM_{2.5}$, but the $SUVA_{254}$ value was in a similar range for all of these types of HULIS. Therefore, caution should be expressed when using just the spectra parameter of $E_{250}/E_{365}$ for the characterization of HULIS.

## 3.4 Fluorescence properties

Fluorescence spectroscopy has been used as a technique for classifying and distinguishing between humic substances of various origins and natures. It has been widely applied to characterize HULIS in atmospheric aerosols (Duarte et al., 2004; Santos et al., 2009; Santos et al., 2012). Figure 2 shows the EEM fluorescence spectra of HULIS in smoke $PM_{2.5}$ from BB and coal combustion, and in ambient $PM_{2.5}$. To avoid concentration effects, the fluorescence spectra were normalized by the WSOC content of HULIS, and are shown here as specific fluorescence intensities (a.u.L/(gC)).

As shown in Figure 2, the four types of primary HULIS in smoke $PM_{2.5}$ have similar fluorescence

features, with two main types of fluorophores at $\lambda_{ex}/\lambda_{em} \approx (245\text{--}255)/(420\text{--}435)$ nm (peak A) and $\lambda_{ex}/\lambda_{em} \approx (265\text{--}290)/(335\text{--}370)$ (peak B). Bands in the same range as peaks A and B have been already identified in the EEM fluorescence spectra of water-soluble organic matter from rainwater (Kieber et al., 2006; Santos et al., 2009; Santos et al., 2012), fogwater (Birdwell and Valsaraj, 2010), and atmospheric

aerosols (Duarte et al., 2004), and have been assigned to fulvic-like and protein-like fluorophores (Kieber et al., 2006), respectively. Our results indicate that the primary HULIS fractions had similar fulvic-like and protein-like organic fractions to atmospheric HULIS (Duarte et al., 2004; Santos et al., 2009; Santos et al., 2012). However some differences in peak A and B were identified between the four types of primary HULIS and the atmospheric HULIS. For peak A, the Ex/Em wavelengths in the EEM

fluorescence spectra of primary HULIS had peaks at longer excitation and emission wavelengths ($\lambda_{ex}/\lambda_{em} \approx (245\text{--}255)/(420\text{--}435)$ nm) than those ($\lambda_{ex}/\lambda_{em} \approx 245/405$ nm) of atmospheric HULIS. This suggests that primary HULIS fractions contain more aromatic structures and condensed unsaturated bond systems, but fewer aliphatic structures (Peuravuori et al., 2002; Duarte et al., 2004; Graber and Rudich, 2006; Krivacsy et al., 2008; Santos et al., 2009). For peak B, the four types of primary HULIS

all had a protein-like fluorescence band at a similar $\lambda ex/\lambda em$ in the EEM fluorescence spectra. In all four types of primary HULIS the band was clearly stronger than the band in ambient HULIS described in this and previous studies (Duarte et al., 2004; Santos et al., 2009; Santos et al., 2012). This finding indicates that these four types of primary HULIS consist of more phenol-like, protein-like, and/or aromatic amino acids than atmospheric HULIS (Coble, 1996; Peuravuori et al., 2002; Duarte et al.,

2004; Kieber et al., 2006).

Compared to the ambient HULIS in this and previous studies (Duarte et al., 2004; Santos et al., 2009; Santos et al., 2012), a distinct band at $\lambda_{ex}/\lambda_{em} \approx 295/405$ nm (peak C), which is generally attributed to humic-like compounds (Coble and Green, 1990; Coble, 1996) was not present in the EEM fluorescence spectra of primary smoke HULIS. This peak C is normally identified in naturally occurring

humic acids and atmospheric HULIS, and has been assigned to marine humic-like compounds

(McKnight et al., 2001; Duarte et al., 2004; Kieber et al., 2006; Santos et al., 2009; Santos et al., 2012). Peak C only occurred in EEMs of atmospheric HULIS, which suggests marine sources were also an important contributor to HULIS in atmospheric aerosols in the Pearl River Delta (PRD) region (Lin et al., 2010a; Fan et al., 2016).

## 3.5 FTIR spectroscopy

The FTIR spectra of the HULIS in smoke $PM_{2.5}$ emitted from the combustion of rice straw, corn straw, pine branch, and coal, and in ambient $PM_{2.5}$ were within the region of 4000–1000 $cm^{-1}$, as shown in Figure 3. All spectra were characterized by a number of absorption bands, exhibiting variable relative intensities, which is typical of humic(-like) materials (Senesi et al., 1989; Havers et al., 1998; Duarte et al., 2007). As shown in Figure 3, the spectra of primary HULIS were similar to those of the atmospheric HULIS and WSOM in this and previous studies (Havers et al., 1998; Krivacsy et al., 2001; Kiss et al., 2002; Duarte et al., 2005; Polidori et al., 2008; Santos et al., 2009; Fan et al., 2013; Duarte et al., 2015). The interpretation of these spectra was based on the assignments given in the literature referred to above for humic(-like) substances, resulting in the major characteristic bands that are marked in Figure 3, with the corresponding assignments listed in Table 5.

As shown in Figure 3, the FTIR spectra of the four types of primary HULIS fractions predominantly exhibit the presence of O-containing functional groups, aliphatic C-H groups, and aromatic ring groups, with the majority of the valence vibrations also being characteristics. The broad and strong band centered at around 3420 $cm^{-1}$ is generally attributed to the OH stretching of phenol, hydroxyl, and carboxyl groups. The strong band near 1720 $cm^{-1}$ is usually assigned to C=O stretching, mainly of carboxyl groups. However, to a lesser extent, ketonic and/or aldehydic C=O groups can also give rise to absorption near this wavenumber, and their contribution should not be neglected. Some bands were also displayed near 1458, 1610 and 1637 $cm^{-1}$, indicating the presence of aromatic groups. These results suggest that primary smoke HULIS are complex compounds, mainly containing aliphatic

chains, carboxylic groups, and aromatic groups. These FTIR spectra features are similar to those of the atmospheric HULIS described in this and other studies (Krivacsy et al., 2001; Duarte et al., 2005; Song et al., 2012; Fan et al., 2013; Duarte et al., 2015).

There were many discriminatory characteristics between primary HULIS and atmospheric HULIS. Relative weaker band at 1710 cm[-1] for primary HULIS than for atmospheric HULIS was observed in Figure 3, indicating the former ones contain fewer carboxyl groups (Song et al., 2012; Fan et al., 2013). The bands at 1458 and 1610 cm[-1], which are generally attributed to the C–C stretching of aromatic rings (Watanabe and Kuwatsuka, 1992; Duarte et al., 2015), are observed in spectra of primary HULIS from direct combustion emissions. However, they were less intense, or even absent in the FTIR spectra of atmospheric HULIS fractions in this and some previous studies (Krivacsy et al., 2001; Duarte et al., 2005; Fan et al., 2013). This indicates that more aromatic rings are present in the primary HULIS from BB and coal combustion than in atmospheric HULIS. In addition, the bands near 1516 and 1115 cm[-1], which are ascribed to the stretching vibrations of aromatic C=C and C–O bonds, were apparent in spectra of primary HULIS from BB, but absent from the atmospheric HULIS in this and some other studies (Santos et al., 2009; Santos et al., 2012; Fan et al., 2013). Because these wavenumber regions are often typically displayed in the spectra of lignin, they can be used as a marker to reflect BB contributions to atmospheric HULIS. For example, this band has been observed in the FTIR spectra of atmospheric HULIS in the autumn and winter seasons, when HULIS is likely to be significantly influenced by BB (Duarte et al., 2005; Duarte et al., 2007). Simultaneously, a relatively strong band at 1045 cm[-1] was found in the spectra of primary HULIS from BB, which is often attributed to the C–O bond stretching of polysaccharides (Havers et al., 1998), and is also a characteristic of BB sources.

There were also some discriminatory differences between the primary HULIS fractions in the minor bands, and in the shape and intensity of the major bands in the 2000–1000 cm[-1] region of FTIR spectra. As shown in Figure 3, relatively sharper and stronger peaks at 1637, 1610, 1458, and 1385 cm[-1] were displayed in the spectra of HULIS fractions in smoke $PM_{2.5}$ from coal combustion than those from

BB. These results indicated that coal smoke HULIS consisted of more aromatic structures. It was noteworthy that additional peaks at 3160 and 1400 cm$^{-1}$ were observed in the spectra of HULIS from the burning of corn straw. They are related to the stretching of C–O and O–H bonds (Watanabe and Kuwatsuka, 1992; Chen et al., 2002), indicating that there were more oxygenated phenolic structures in the HULIS from corn straw burning. This is consistent with the results derived from the elemental analysis, in which a higher O/C molar ratio was obtained for HULIS from corn straw burning. One distinct feature of the primary HULIS from BB was the occurrence of one sharper and stronger peak at 1516 cm$^{-1}$ on the FTIR spectra. This band is generally assigned to stretching vibrations of aromatic C=C bonds and C–O bonds, and is often observed in the spectra of compounds derived from lignin (Watanabe and Kuwatsuka, 1992; Santos et al., 2000; Duarte et al., 2003). Consequently, primary HULIS from BB displayed the characteristic of containing a lignin-like structure in its molecules, which can be seen as an important indicator of a BB source.

### 3.6 $^1$H-NMR spectroscopy

A critical review on the application of $^1$H NMR spectroscopy on WSOM in atmospheric aerosols has been presented, in which $^1$H NMR was demonstrated to be an important and novel tool to characterize WSOM, which can not only provide deep insight into the structural characteristics of them but also reflect their sources (Duarte and Duarte, 2015). In this work, $^1$H NMR was applied to characterize the primary HULIS from BB and coal combustion and atmospheric HULIS. Figure 4 shows the $^1$H NMR spectra of the four types of primary HULIS in smoke PM$_{2.5}$ emitted from the combustion of rice straw (a), corn straw (b), pine branch (c), and lignite coal (d), and atmospheric HULIS in ambient PM$_{2.5}$ (e). The four types of primary HULIS displayed similar spectra to the atmospheric HULIS spectra in this study, which were also comparable to $^1$H NMR spectra of HULIS and/or WSOM in fog (Decesari et al., 2000), cloud (Decesari et al., 2005), rain water (Santos et al., 2009, 2012), biomass burning aerosols (Graham et al., 2002) and urban/rural aerosols (Decesari et al.,

2007; Ziemba et al., 2011;Song et al., 2012; Fan et al., 2013; Lopes et al., 2015).

As shown in Figure 4, compared with the atmospheric HULIS, more distinct sharp signals of organic species could be seen in the [1]H NMR spectra of primary HULIS. According to the results of Fan et al. (2012), some low molecular weight organic compounds (e.g., suberic, 3,5-dihydroxybenzoic, and phthalic acids) are generally present in isolated HULIS fractions. Thus, these sharp peaks in the [1]H NMR spectra of primary HULIS can be ascribed to low molecular weight organic species in the smoke $PM_{2.5}$. The relatively few and/or weak sharp peaks in the [1]H NMR spectra of atmospheric HULIS may be the result of low molecular weight organic compounds that have been removed by oxidation or transformed to HULIS. Among these sharp signals, a limited number of resonances could be attributed to specific organic species by comparison with previous studies (Decesari et al., 2000, 2001; Suzuki et al., 2001; Matta et al., 2003; Cavalli et al., 2006; Chalbot et al., 2014, 2016;Lopes et al., 2015). These sharp signals included low molecular weight formate (δ8.44 ppm), terephthalic acid (δ8.01 and 8.12 ppm), phthalic acid (δ7.45–7.47 and 7.58 ppm), glucose (δ3.88–3.91 and 3.81–3.85 ppm), fructose (δ3.79–3.84 ppm), trimethylamine (δ2.71 and 2.89 ppm), dimethylamine (δ2.72 ppm), monomethylamine (δ2.55 ppm). It is worth noting that all BB-derived HULIS present more sharp glucose and fructose resonances than atmospheric HULIS in [1]H NMR spectra, but they were absent for coal combustion derived HULIS. On the other hand, coal combustion derived HULIS contain more sharp resonances of terephthalic acid and phthalic acid than atmospheric HULIS, but they were absent for BB-derived HULIS. Moreover, both BB and coal combustion derived HULIS exhibit many sharp signals between 6.5–8.5 ppm, which could ascribed to aromatic structures, such as substituted phenols and alkylbenzenes (around 6.6–7.0 ppm), benzoic acids, esters, and nitroaromatics (Suzuki et al., 2001; Chalbot et al., 2014).

Although some sharp peaks were identified above, most of the signals in the spectra of all HULIS fractions appeared as a continuous unresolved distribution. This suggests that HULIS consists of a complex mixture of organic substances (Samburova et al., 2007; Song et al., 2012; Fan et al., 2013;

Lopes et al., 2015). The integrated [1]H NMR signal over specific ranges of chemical shift has been used previously to quantify the contribution of organic functional groups in HULIS from urban/rural aerosols (Song et al., 2012; Fan et al., 2013; Lopes et al., 2015) and rainwater (Miller et al., 2009; Santos et al., 2009, 2012). Accordingly, four main regions of chemical shifts were assigned and integrated in the spectra: $\delta H$ = 0.6–2.0 ppm (aliphatic protons in alkyl chains, [H–C]); $\delta H$ = 2.0–3.2 ppm (aliphatic protons attached to C atoms adjacent to a carbonyl or aromatic group [H–C–C=]); $\delta H$ = 3.4–4.4 ppm (protons on C atoms singly bound to O or other heteroatoms, indicative of protons associated with carbohydrates, ethers, or esters, [H–C–O]); and $\delta H$ = 6.5–8.5 ppm (aromatic protons, [Ar–H]). It is worth noting that a few distinct weak peaks between 9 and 10 ppm were observed in the primary HULIS in fresh soot particles, and can be attributed to aldehydic protons, [H–C=O] (Ziemba et al., 2011). However, they were absent in the atmospheric HULIS or WSOM in this and other studies (Decesari et al., 2005; Cavalli et al., 2006; Decesari et al., 2007; Samburova et al., 2007; Song et al., 2012; Fan et al., 2013; Lopes et al., 2015). This region only accounts for a minor fraction (<2%), and therefore was not considered further.

Table 6 shows the distribution of the four types of protons described above, estimated from the area of the observed [1]H NMR regions for different HULIS samples. The four types of primary HULIS in fresh smoke particles contained a relatively high content of H–C (27%–45%), H–C–C= (22%–40%), and Ar–H (19%–31%) groups, and a relatively low content of the H–C–O group (2-19%). These four functional groups have also been observed in the [1]H NMR spectra of atmospheric HULIS in this and other studies, but the relative distribution of these four functional groups are different. The atmospheric HULIS in ambient aerosols from this work and other studies (Song et al., 2012; Fan et al., 2013; Lopes et al., 2015) and in rainwater (Miller et al., 2009; Santos et al., 2009, 2012) were all characterized by the highest content of [H–C] (37–60%), moderate contents of [H–C–C=] (20–37%) and [H–C–O] (10–24%), and the lowest content of [Ar–H] (1–12%). It was noted that the relative content of [Ar–H] groups (19–31%) in primary HULIS was significantly higher than that in atmospheric HULIS. This

suggests that primary HULIS contained more aromatic structures, which is consistent with the elemental composition, UV-vis spectra, EEM fluorescence spectra, and FTIR spectra results. This result is also consistent with the observations of more aromatic protons in HULIS in colder seasons ascribed to BB influence (Song et al., 2012; Lopes et al., 2015).

Although similarities existed, some differences on functional groups distributions were also observed between the primary HULIS from BB and coal combustion. As shown in Table 6, there was a relatively higher content of [H–C–C=] (40%) and [Ar–H] (31%) in primary HULIS from coal combustion than from BB, indicating that the primary HULIS in coal smoke contained more unsaturated aliphatic (i.e., carbonyl groups (C=O)) and aromatic structural groups. In addition, the content of [H–C–O] in HULIS from coal combustion was only 2%, which was significantly lower than in HULIS from BB. As described above, the [H–C–O] group was assigned to protons associated with carbohydrates and ethers. Therefore, these differences could be ascribed to the fact that the HULIS from BB contained a relatively high content of carbohydrate derived compounds. The lower molar O/C and mass OM/OC ratios, and lower intensity of peaks at 1516 and 1045 cm$^{-1}$ and higher intensity of peaks at 1637, 1610, 1458, and 1385 cm$^{-1}$ in the FTIR spectra of HULIS from coal burning all support this finding. For the HULIS from BB, a relatively high content of [H–C] and low content of [Ar–H] were observed for HULIS from pine branch combustion when compared to the HULIS from rice straw and corn straw combustion. These results suggest that the primary HULIS from pine branch combustion contained more aliphatic protons and fewer aromatic protons than the HULIS from rice straw and corn straw combustion.

## 3.7 Light absorption properties

Recently, the organic aerosols not only black carbon or elemental carbon have been drawn more and more attentions for their light absorption effects. WSOM are important fractions in organic aerosols, which have been documented to have strong light absorbing properties (Chen and Bond, 2010;

Hecobian et al., 2010; Cheng et al., 2011; Liu et al., 2013; Zhang et al., 2013; Du et al., 2014; Kirillova et al., 2014; Yan et al., 2015; Cheng et al., 2016; Kim et al., 2016; Park and Yu, 2016). The AAE and MAE$_{365}$ have been used to reflect the light absorbing properties of water soluble brown carbon (refer to WSOM) in above studies. However, the investigation on light absorbing properties of primary HULIS from direct combustion emissions is very limited. As an important component of WSOM, the observation on light absorbing properties of HULIS is valuable for further understanding on their environment effects.

Light absorption of primary HULIS in this study increased sharply towards shorter wavelengths (not shown), which is characteristic for brown carbon spectra (Du et al., 2014; Park and Yu, 2016). The AAE of HULIS fitted between wavelengths of 330 to 400 nm for rice straw, corn straw, pine branch, and coal combustion smoke PM$_{2.5}$ were 8.2 $\pm$ 0.6, 6.7 $\pm$ 0.6, 6.7 $\pm$ 0.7 and 13.6 $\pm$ 0.2, respectively (Table 7). The AAE values of primary HULIS from biomass burning are comparable to those of atmospheric HULIS from urban aerosols in this study (7.0 $\pm$ 0.2) and from the Amazon biomass burning aerosols in earlier research (6.4–6.8) (Hoffer et al., 2006). Because of limited observation on AAE of HULIS, the comparisons between HULIS and WSOM were conducted. As seen in Table 7, primary HULIS and WSOM from same smoke PM$_{2.5}$ almost present the same AAE values. Those AAE values of HULIS were also in the range of those of primary WSOM from BB (7.4-17.8) and atmospheric WSOM from urban/rural aerosols (5.3–8.3) reported in previous studies (Chen and Bond, 2010; Hecobian et al., 2010; Cheng et al., 2011; Liu et al., 2013; Zhang et al., 2013; Du et al., 2014; Kirillova et al., 2014; Yan et al., 2015; Cheng et al., 2016; Kim et al., 2016; Park and Yu, 2016). It is worth noting that the AAE of HULIS from coal combustion is observed as 13.6, which is significantly higher than those of primary HULIS or WSOM from BB in this study and in Park and Yu (2016). They are also substantially higher than those of atmospheric WSOM from ambient aerosols in here and previous studies (Hecobian et al., 2010; Cheng et al., 2011; Liu et al., 2013; Zhang et al., 2013; Du et al., 2014; Kirillova et al., 2014; Yan et al., 2015; Cheng et al., 2016; Kim et al., 2016; Park and Yu, 2016).

The mass absorption efficiency ($MAE_{365}$), which characterizes the efficiency of absorbing solar energy by per DOC of HULIS, was also investigated in this study. As observed in Table 7, the $MAE_{365}$ of primary HULIS were 1.54 ±0.30, 2.09 ±0.41, 0.97 ±0.22 and 0.63 ±0.03 $m^2/gC$ for rice straw, corn straw, pine branch and coal smoke emissions, respectively. It is obvious that the $MAE_{365}$ of HULIS were higher than those of the corresponding WSOM, suggesting a stronger absorbing ability of HULIS. Moreover, these primary HULIS $MAE_{365}$ values seems to be comparable to ambient WSOM $MAE_{365}$ (0.13-1.79 $m^2/gC$) in previous studies (Hecobian et al., 2010; Cheng et al., 2011; Liu et al., 2013; Zhang et al., 2013; Du et al., 2014; Kirillova et al., 2014; Yan et al., 2015; Cheng et al., 2016; Kim et al., 2016; Park and Yu, 2016). For the four primary HULIS, it is noteworthy that $MAE_{365}$ of primary HULIS from BB was typically ~1.5–3 times higher than from coal combustion. It suggests that primary HULIS from BB contain more light absorbing chromophores than HULIS from coal combustion, which could significantly affect the light absorbing abilities of organic aerosols.

## 4. Conclusions

In this work, the primary HULIS fractions in smoke $PM_{2.5}$ emitted from the combustion of biomass materials and fossil fuels were isolated and comprehensively characterized by various analytical methods, including TOC analysis, elemental analysis, UV-vis, EEM fluorescence, FTIR, and $^1$H NMR spectroscopy. The main conclusions were:

(1) The HULIS fractions were important components of smoke $PM_{2.5}$ obtained from the combustion of biomass materials and coal, and accounted for 5.3%− 23.4% of PM, 5.2%− 21.7% of TC, and 45.5%−66.1% of WSOC, respectively. These results indicate that BB and coal combustion are all important sources of HULIS in atmospheric aerosols. However, the HULIS fractions in diesel soot only accounted for 0.8% of soot particles, suggesting that the primary vehicular exhaust source for atmospheric HULIS may be negligible.

(2) The primary and atmospheric HULIS were very similar in many aspects. At first, they had

similar chemical compositions, in which C and O were the dominant elements. Moreover, many similarities in chemical properties and structures were also detected. For example, the UV-vis spectra of primary HULIS and atmospheric HULIS were all characterized by features that indicated the absorbance decreased as the wavelength increased. The two main peaks assigned to fulvic-like (peak A) and protein-like (peak B) fluorophores were both observed in the EEM spectra of primary and atmospheric HULIS. In the case of the [1]H NMR analysis, four main regions of chemical shifts, assigned to [H–C], [H–C–C=], [H–C–O] and [Ar–H] in the primary HULIS, were also found in atmospheric HULIS. The AAE and $MAE_{365}$ of the BB derived primary HULIS were similar to those of atmospheric HULIS and/or WSOM.

(3) There were also some differences identified between primary and atmospheric HULIS. At first, the O/C atomic ratios of coal combustion derived HULIS was significantly lower than the ratio for atmospheric HULIS samples. Moreover, the primary HULIS contained more polycyclic aromatic and conjugated compounds than atmospheric HULIS, as consistently revealed by the UV-vis, EEM fluorescence, FTIR, and [1]H NMR spectroscopy analysis. For example, primary HULIS exhibit relative higher absorbance in shorter wavelength regions than atmospheric HULIS in the UV-vis spectra. The relative content of the [Ar–H] group in primary HULIS was significantly higher than in atmospheric HULIS, as shown by the [1]H NMR analysis. In addition, many sharp signals of organic species ascribed to low molecular weight aromatic organic compounds were observed in the [1]H NMR spectra of primary HULIS, but they were not as abundant in the [1]H NMR spectra of atmospheric HULIS.

(4) Some distinct features were also identified among the four types of primary HULIS. For example, the BB HULIS contain a relatively higher content of O-containing components than HULIS from coal combustion as revealed by elemental analysis. In addition, the results from the FTIR and [1]H NMR spectroscopy indicated that the primary HULIS from BB contained relative high contents of lignin-like and carbohydrate derived structures, while the primary HULIS from coal combustion contained relatively high levels of aromatic structures. The $MAE_{365}$ of BB HULIS are 0.97-2.09 $m^2/g$,

which are higher than that of coal combustion HULIS, suggesting the former one own stronger light absorption properties. For the three types of BB HULISs, a relatively higher content of [H–C] and lower content of [Ar–H] were observed for HULIS in pine branch smoke than HULIS in the rice straw and corn straw smokes, suggesting that the primary HULIS from pine combustion contained more aliphatic protons and less aromatic protons.

## 5. Implications

As a significant fraction of water soluble organic matter, HULIS has been widely studied in recent years. However, the studies of primary HULIS directly emitted from combustion processes with respect to amount and chemical properties are still limited. This work is a comprehensive study for the primary HULIS from direct combustion of rice straw, corn straw, pine branch, coal and diesel fuels. The results confirmed that combustion processes including BB and coal combustion are significant sources of atmospheric HULIS, but the vehicular exhaust source for primary HULIS may be negligible. It's noted that this is the first time that coal combustion was identified as an important source of primary HULIS. Moreover, the chemical properties, and structures of primary HULIS from combustion process were really comprehensively characterized. Many similarities of chemical aspects were observed between primary HULIS and atmospheric HULIS, but some distinct features were also identified for the primary HULIS. These comprehensive characterizations of primary HULIS in smoke $PM_{2.5}$ are very helpful to the better understanding of the chemical natures of primary HULIS from direct combustion emissions and their contribution to atmospheric HULIS. Nevertheless, some questions are still remained and more efforts should be made in the future: 1) the emission factors and chemical characteristics of primary HULIS formed from combustion of more types of biomass materials, coals, etc.; 2) the emission factors and chemical characteristics of primary HULIS formed under controlled combustion conditions (e.g., flaming or smoldering burns, combustion temperature and air dilution ratio); 3) the aging process of primary HULIS in atmospheric environment.

*Acknowledgments.* The work was supported by the Natural Science Foundation of China (Grants 41390242, 41473104, and 41173110), and the Foundation for Leading Talents from Guangdong Province Government. The authors want to thank Dr. Shuiping Wu from Xiamen University for offering the possibility to prepare and collect smoke $PM_{2.5}$ samples in their own resuspension chamber in State Key Laboratory of Marine Environmental Science.

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

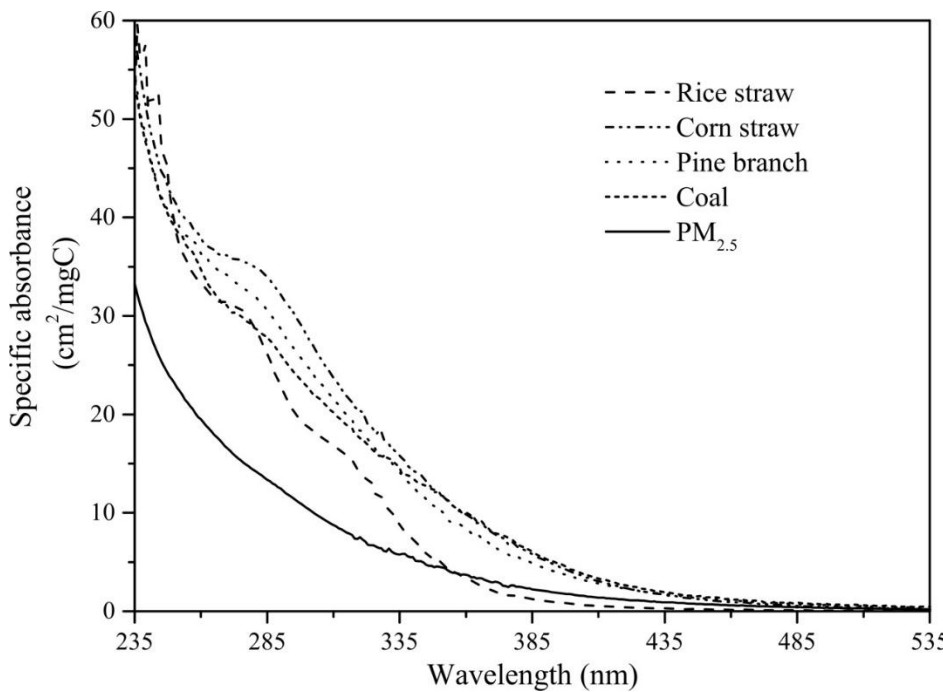

**Figure 1.** The normalized UV-vis spectra with TOC (units: $cm^2/mgC$) of HULIS in smoke $PM_{2.5}$ from combustion of rice straw, corn straw, pine branch and coal, and in ambient $PM_{2.5}$.

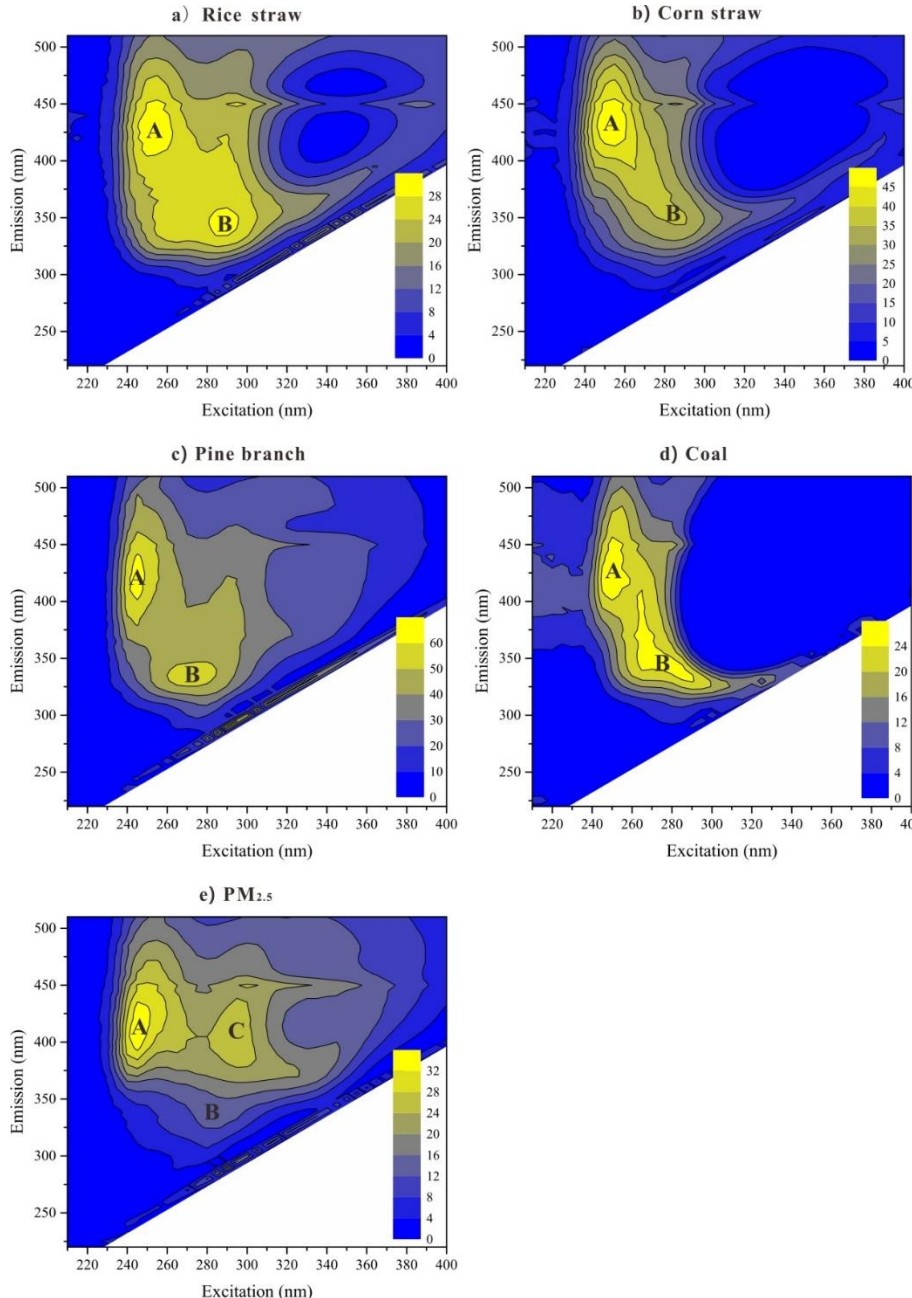

**Figure 2.** EEM spectra of HULIS in smoke $PM_{2.5}$ from combustion of rice straw (a), corn straw (b), pine branch (c), coal (d), and in ambient $PM_{2.5}$ (e), presented as specific intensity (a.u. L/(g C)).

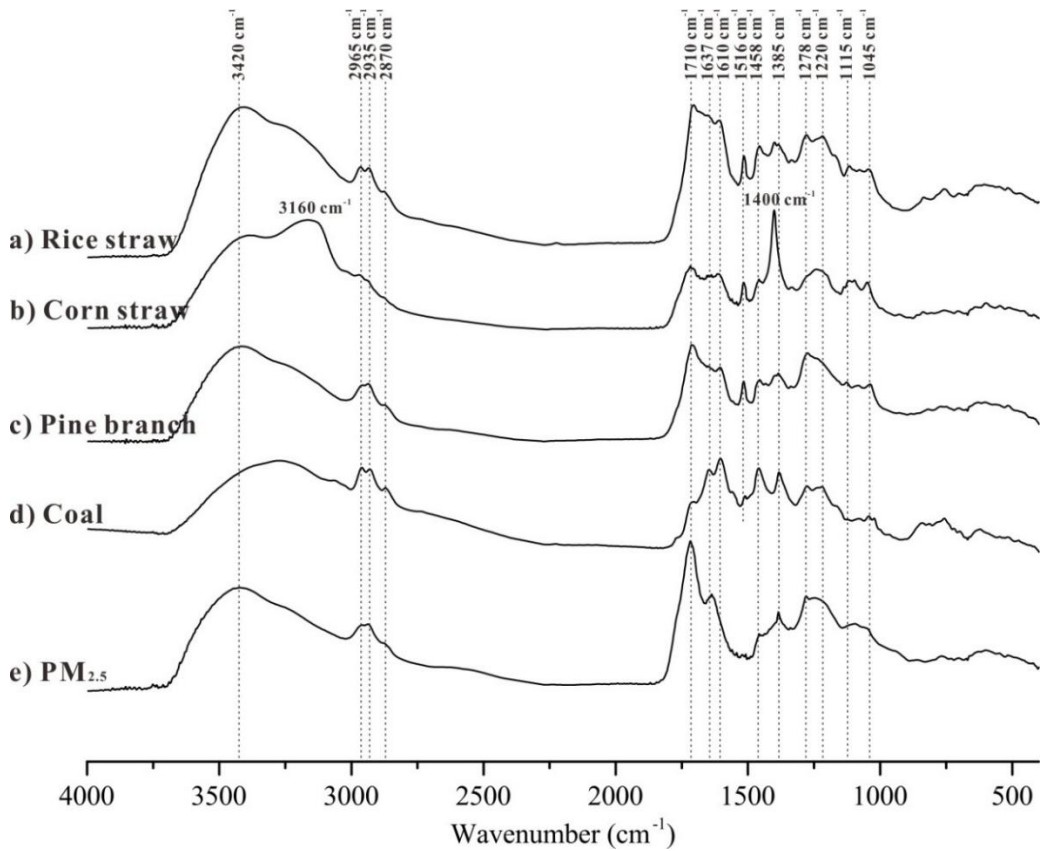

**Figure 3.** FTIR spectra of primary HULIS in smoke PM$_{2.5}$ from combustion of rice straw (a), corn straw (b), pine branch (c), coal (d), and in ambient PM$_{2.5}$ (e).

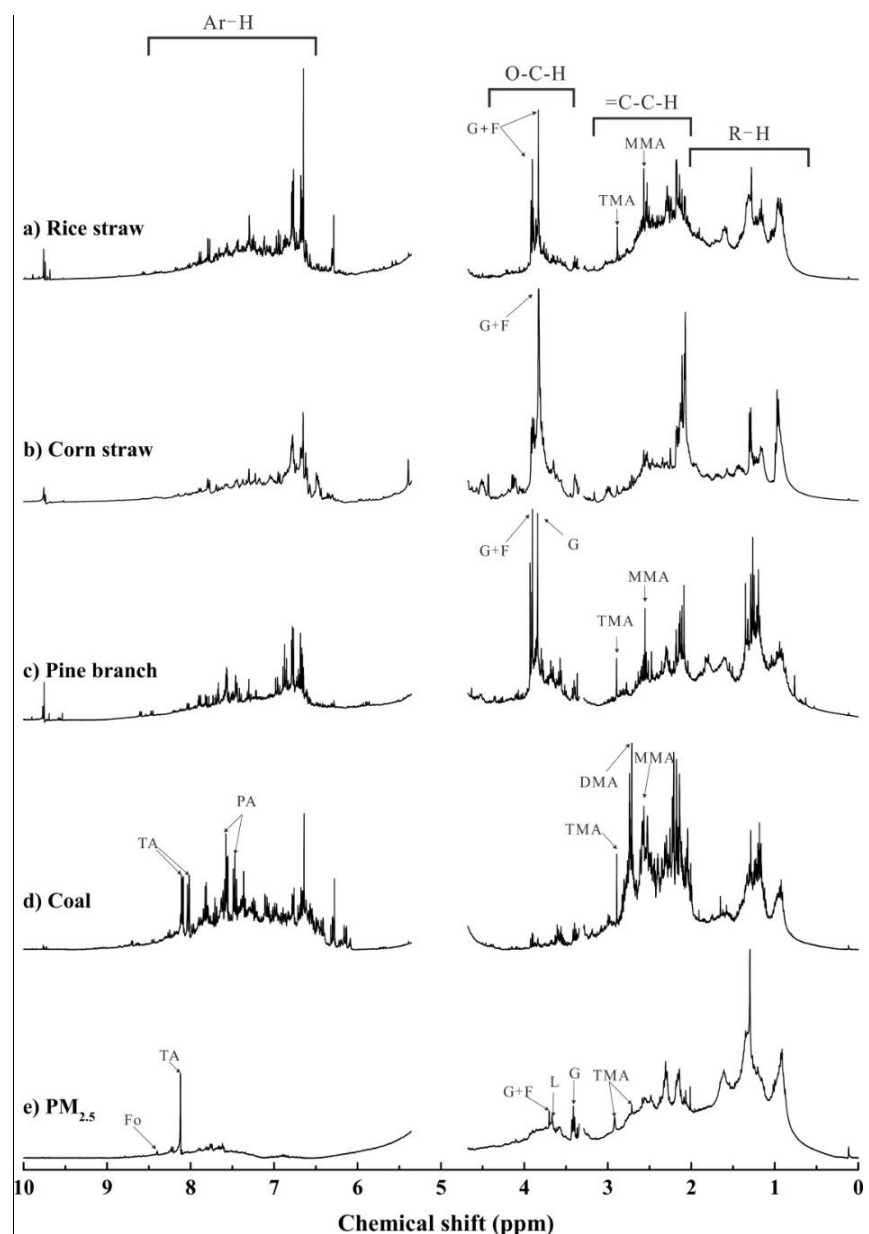

**Figure 4.** $^1$H NMR spectra of HULIS in smoke PM$_{2.5}$ from combustion of rice straw (a), corn straw (b), pine branch (c), coal (d), and in ambient PM$_{2.5}$ (e). The segments from 3.30 to 3.35 ppm and 4.80 to 5.40 ppm were removed from all NMR spectra due to MeOH and H$_2$O residues. The peaks were assigned to specific compounds as follows: formate (Fo), terephthalic acid (TA), phthalic acid (PA), glucose (G), fructose (F), trimethylamine (TMA), dimethylamine (DMA), monomethylamine (MMA).

Table 1. Proximate analysis of three biomass materials and two fossil fuels (wt%) (n=4)

| Materials | Moisture (%) | C (%) | H (%) | N (%) | S (%) | O (%) |
|---|---|---|---|---|---|---|
| Rice straw | 5.8 ±0.5 | 36.0 ±1.0 | 5.4 ±0.5 | 0.6 ±0.1 | 0.1 ±0.0 | 37.1 ±1.2 |
| Corn straw | 7.4 ±0.8 | 38.7 ±1.7 | 6.6 ±0.2 | 0.5 ±0.1 | 0.2 ±0.1 | 44.2 ±0.5 |
| Pine branch | 7.6 ±0.7 | 44.7 ±0.3 | 7.2 ±0.0 | 0.2 ±0.1 | 0.1 ±0.0 | 45.0 ±0.4 |
| Coal | 1.6 ±0.2 | 72.6 ±0.6 | 4.0 ±0.4 | 0.9 ±0.0 | 1.7 ±0.1 | 8.2 ±0.3 |
| Diesel soot | - | 86.5 ±0.2 | 1.4 ±0.1 | 0.4 ±0.0 | 0.8 ±0.1 | 9.1 ±0.2 |

**Table 2.** The contributions of HULIS to particular matters (PM), total carbon (TC), water soluble organic matter (WSOM) in smoke PM$_{2.5}$ emitted from combustion of biomass materials and fossil fuels, and in ambient PM$_{2.5}$.

| Samples | Types | Isolation methods | HULIS-C/PM (µgC/µg, %) | HULIS/PM (%) | HULIS-C/OC (%) | HULIS-C/TC (%) | HULIS/WSOM (TOC, %) | HULIS/WSOM (UV250, %) | References |
|---|---|---|---|---|---|---|---|---|---|
| Rice straw | Smoke PM$_{2.5}$ (n=5)[a] | HLB | 13.5 ±3.1 | 23.4 ±5.5 [b] | - | 21.7 ±4.0 | 66.1 ±2.4 | 79.5 ±1.5 | |
| Corn straw | Smoke PM$_{2.5}$ (n=5) | HLB | 5.9 ±4.0 | 11.2 ±7.5 [b] | - | 14.7 ±6.9 | 59.2 ±2.4 | 75.4 ±2.9 | |
| Pine branch | Smoke PM$_{2.5}$ (n=5) | HLB | 6.4 ±2.2 | 11.4 ±3.8 [b] | - | 8.0 ±2.9 | 56.9 ±3.1 | 68.1 ±5.5 | Present work |
| Coal | Smoke PM$_{2.5}$ (n=5) | HLB | 3.5 ±0.3 | 5.3 ±0.4 [b] | - | 5.2 ±0.3 | 45.5 ±2.1 | 64.4 ±3.9 | |
| Diesel soot | SRM 2975 (n=5) | HLB | 0.5 ±0.01 | 0.8 ±0.02 [c] | | 0.7 ±0.0 | 62.3 ±3.7 | 58.3 ±3.5 | |
| Urban aerosols | PM$_{2.5}$ (n=5) | HLB | 5.2 ±0.4 | 10.7 ±0.8 [b] | 26.8 ±3.3 | 22.6 ±3.7 | 60.7 ±1.0 | 70.7 ±2.1 | |
| Leaf | Smoke PM$_{10}$ | C18-SAX | 18.5 - 21.2 | - | 33.0 - 34.5 | 27.8 - 31.3 | - | - | Schmidl et al. (2008a) |
| Wood | Smoke PM$_{10}$ | C18-SAX | 0.6 - 5.8 | - | 1.0 - 12.0 | 0.9 - 9.2 | - | - | Schmidl et al. (2008b) |
| Wood | Smoke PM$_{10}$ | C18-SAX | 1.5 - 2.4 | - | 3.5 - 11.5 | 2.8 - 5.3 | - | - | Goncalves et al. (2010) |
| Sugarcane | Smoke PM$_{2.5}$ | ELSD | - | 7.6 | 14.3 | - | 33 | - | Lin et al. (2010b) |
| Rice straw | Smoke PM$_{2.5}$ | ELSD | - | 12.4 | 14.7 | - | 30 | - | Lin et al. (2010b) |
| Rice straw | Smoke PM$_{2.5}$ | HLB | 15 ±1 | 29.5 ±2.0 | 26 ±3 | 24.2 | 63 ±5 | | Park and Yu (2016) |
| Pine needles | Smoke PM$_{2.5}$ | HLB | 8 ±3 | 15.3 ±3.1 | 15 ±4 | 14.9 | 36 ±8 | | Park and Yu (2016) |
| Sesame stems | Smoke PM$_{2.5}$ | HLB | 13 ±4 | 25.8 ±4.0 | 29 ±8 | 28.3 | 51 ±8 | | Park and Yu (2016) |
| Vehicular exhaust | PM$_{2.5}$ | DEAE | 0.6 | - | 2.9 | 1.0 | 18.4 | - | I.El Haddad et al. (2009) |
| Vehicular exhaust | PM$_{10}$ | DEAE | 0.8 | - | 3.4 | 1.3 | 20.7 | - | I.El Haddad et al. (2009) |
| Roadway site aerosols | PM$_{2.5}$ | XAD7 HP | 5.2 | - | 34.5 | 26.6 | 59.8 | - | Park et al. (2013) |
| Marine aerosol | PM$_{10}$ | HLB | - | - | - | 12 | 19 | - | Krivacsy et al. (2008) |
| Rural aerosols | PM$_{1.5}$ | HLB | - | - | | 39 | 57 | - | Kiss et al. (2002) |
| Rural aerosols | PM$_{2.5}$ | XAD-8 | 4.3 | - | 23.2 | 19.5 | 51.9 | - | Duarte et al. (2007) |
| Urban aerosol | PM$_{2.5}$ | ELSD | 6.0 | 11.7 | 29.5 | - | 60 | - | Lin et al. (2010b) |
| Urban aerosol | PM$_{2.5}$ | HLB | - | 6.7 | 18.3 | 10.6 | 62 | - | Salma et al. (2007) |
| Urban aerosol | PM$_{2.5}$ | HLB | - | - | 26.6 - 28.9 | 18.4 - 20.8 | 45.8 - 49.7 | - | Salma et al. (2008) |
| Urban aerosol | PM$_{2.5}$ | XAD7HP | - | - | 35.4 | 27.6 | 63.0 | - | Park et al. (2012) |
| Urban aerosol (dust) | PM$_{2.5}$ | XAD7HP | 6.2 | - | 31.4 | 24.6 | 71.9 | - | Park and Cho (2013) |

| Urban aerosols | PM$_{2.5}$ | ENVI-18 | 2.8 ±1.3 | 5.4 ±2.7 | 16.9 ±4.9 | 13.0 ±4.5 | 49.5 ±5.9 | 68.3 ±4.7 | Fan et al. (2016) |

[a] indicating the number of replicates of samples

[b] HULIS mass are calculated by OM/OC ratios obtained by elemental analysis listed in Table 3.

[c] HULIS mass is calculated by the OM/OC ratio (1.51) of primary HULIS in coal smoke PM$_{2.5}$.

**Table 3.** Elemental composition and molar ratios of HULIS in smoke $PM_{2.5}$ from combustion of rice straw, corn straw, pine branch and coal and in ambient aerosols.

| Samples | Types | Isolation methods | Elemental composition (%) | | | | Molar ratios | | | OM/OC[b] | References |
|---|---|---|---|---|---|---|---|---|---|---|---|
| | | | N | C | H | O[a] | H/C | O/C | N/C | | |
| Rice straw | Smoke $PM_{2.5}$ | HLB | 4.1 | 57.4 | 5.5 | 33.0 | 1.15 | 0.43 | 0.06 | 1.74 | |
| Corn straw | Smoke $PM_{2.5}$ | HLB | 2.5 | 52.9 | 6.3 | 38.3 | 1.43 | 0.54 | 0.04 | 1.89 | |
| Pine branch | Smoke $PM_{2.5}$ | HLB | 2.0 | 56.7 | 6.6 | 34.7 | 1.40 | 0.46 | 0.03 | 1.77 | Present work |
| Coal | Smoke $PM_{2.5}$ | HLB | 3.3 | 66.1 | 7.1 | 23.6 | 1.28 | 0.27 | 0.04 | 1.51 | |
| Urban aerosols | $PM_{2.5}$ | HLB | 2.9 | 48.5 | 6.4 | 42.1 | 1.59 | 0.65 | 0.05 | 2.06 | |
| Alpine aerosol | $PM_{2.5}$ | C-18 | 2.5 | 52 | 6.7 | 38 | 1.53 | 0.55 | 0.04 | 1.91 | Krivacsy et al. (2001) |
| Rural aerosol | $PM_{1.5}$ | HLB | 2.5 | 52 | 6.2 | 39 | 1.43 | 0.58 | 0.04 | 1.93 | Kiss et al. (2002) |
| Rural aerosol | $PM_{2.5}$ | XAD-8 | 2.1-3.8 | 51-58 | 5.6-6.5 | 32-37 | 1.21-1.42 | 0.41-0.55 | 0.03-0.06 | 1.71-1.95 | Duarte et al. (2007) |
| Urban aerosol | $PM_{2.5}$ | HLB | 3.1 | 55 | 7 | 35 | 1.49 | 0.47 | 0.05 | 1.82 | Salma et al. (2007) |
| Urban aerosol | TSP | HLB | 2.0-3.9 | 43-53 | 4.4-6.9 | 38-44 | 1.07-1.9 | 0.55-0.76 | 0.03-0.07 | 1.89-2.28 | Song et al. (2012) |
| Urban aerosol | $PM_{2.5}$ | HLB | 3.1 | 54 | 5.9 | 38 | 1.31 | 0.53 | 0.05 | 1.86 | Fan et al. (2013) |
| Urban aerosol | $PM_{2.5}$ | DAX-8 | 2.3-2.6 | 60.4-65.7 | 5.61-6.08 | 25.9-31.2 | 1.01-1.15 | 0.30-0.39 | 0.03-0.04 | 1.5-1.7 | Duarte et al. (2015) |

[a] Calculated as $O(\%) = 100\% - C(\%) - H(\%) - N(\%)$.

[b] Calculated as OM to OC mass ratios

**Table 4.** Absorptivity ratio between 250 nm and 365 nm ($E_{250}/E_{365}$), and the specific UV absorbance at 254 nm ($SUVA_{254}$) of HULIS in smoke $PM_{2.5}$ from combustion of rice straw, corn straw, pine branch and coal, and in ambient $PM_{2.5}$.

| Samples | Type | Isolation methods | $E_{250}/E_{365}$ | $SUVA_{254}$ | References |
|---|---|---|---|---|---|
| Rice straw | Smoke $PM_{2.5}$ | HLB | 5.8±0.5 | 3.7±0.5 | |
| Corn straw | Smoke $PM_{2.5}$ | HLB | 4.5±0.2 | 3.9±0.7 | |
| Pine branch | Smoke $PM_{2.5}$ | HLB | 4.4±0.3 | 3.7±0.4 | Present work |
| Coal | Smoke $PM_{2.5}$ | HLB | 14.7±0.7 | 3.7±0.1 | |
| Urban aerosols (Fall) | $PM_{2.5}$ | HLB | 7.2±0.3 | 2.5±0.1 | |
| SRFA | - | - | 4.86 | 3.94 | |
| Urban aerosols | $PM_{2.5}$ | ENVI-18, HLB, XAD-8 and DEAE | 4.7-5.2 | 2.6-4.6 | Fan et al. (2012) |
| Rural aerosols | $PM_{2.5}$ | ENVI-18, HLB, XAD-8 and DEAE | 5.5-6.2 | 2.7-2.8 | |
| Urban aerosols | Annual $PM_{2.5}$ | ENVI-18 | 5.9±0.9 | 3.2±0.5 | Fan et al. (2016) |
| Urban | Summer $PM_{10}$ | HLB | 7.3, 9.7 | | Krivacsy et al. |
| Urban | Winter $PM_{10}$ | HLB | 5.6, 5.7 | | (2008) |
| Rural aerosols | $PM_{1.5}$ | HLB | 8.0 | - | Kiss et al. (2002) |
| Urban aerosols | Cold season $PM_{2.5}$ | DEAE | 3.1-3.5 | - | |
| Urban aerosols | Summer $PM_{2.5}$ | DEAE | 4.6-5.9 | - | Baduel et al. |
| Urban aerosols | Mid season $PM_{2.5}$ | DEAE | 3.4-3.6 | - | (2010) |
| Biomass burning background aerosols | Urban $PM_{2.5}$ | DEAE | 2.9±0.2 | - | |
| Rural aerosols | Summer $PM_{2.5}$ | XAD-8 | 8.9 | - | |
| Rural aerosols | Autumn $PM_{2.5}$ | XAD-8 | 6.1 | - | Duarte and Duarte (2005) |
| Urban/oceanic aerosols | $PM_{2.5}$ | XAD-8 | 5.8 | - | |

**Table 5.** Major band assignments for FT-IR spectra of HULIS in smoke $PM_{2.5}$ from biomass burning and coal combustion, and in ambient aerosols.

| Wavenumber ($cm^{-1}$) | Band assignments |
| --- | --- |
| 3420 | Stretching vibration of OH |
| 2850-2980 | Stretching vibrations of aliphatic C-H |
| 1710 | Stretching mainly of carboxyl-C and traces of ketones and esters C=O |
| 1637 | Stretching mainly of aromatic C=C and ketones, quinones and amides C=O |
| 1610 | Stretching vibration of aromatic rings |
| 1516 | Stretching vibrations of aromatic C=C |
| 1458 | Deformation of $CH_2$ and $CH_3$ bending and stretching vibration of aromatic rings |
| 1385 | Deformation of aliphatic C-H (some C-O stretching of phenolic OH) |
| 1278 | Stretching of aromatic C-O and phenolic OH |
| 1220 | Stretching vibrations of C-O and deformation of carboxylic O-H |
| 1115 | Stretching of ring breathing C-O |
| 1045 | Stretching of polysaccharide C-O or deformation of aromatic C-H |

**Table 6.** The proton species and corresponding content percentage of HULIS in smoke $PM_{2.5}$ from combustion of rice straw, corn straw, pine branch and coal, and in ambient $PM_{2.5}$.

| Sources | Types/sites | Isolation methods | H-C [a] (0.6-2.0 ppm) | H-C-C= (2.0-3.2 ppm) | H-C-O (3.4-4.4 ppm) | Ar-H (6.5-8.5 ppm) | References |
|---|---|---|---|---|---|---|---|
| Rice straw | Burning emissions | HLB | 34 [b] | 30 | 10 | 27 | |
| Corn straw | Burning emissions | HLB | 31 | 26 | 19 | 24 | |
| Pine branch | Burning emissions | HLB | 45 | 22 | 14 | 19 | Present work |
| Coal | Combustion emissions | HLB | 27 | 40 | 2 | 31 | |
| $PM_{2.5}$ | Urban aerosol | HLB | 51 | 31 | 11 | 6 | |
| Rain water | Terrestrial/marine influenced | C18 | 51.4 (0.5-1.9) | 32.1 (1.9-3.3) | 15.0 (3.4-4.5) | 1.4 (6.5-9.0) | Miller et al. (2009) |
| Rain water | Urban | DAX-8 | 45-51 (0.6-1.8) | 22-29 (1.8-3.2) | 17-24 (3.2-4.1) | 2-7 (6.5-8.5) | Santos et al. (2012) |
| TSP | Urban aerosol | HLB | 53.9-59.8 (0.7-2.0) | 20.1-26.6 (2.0-3.2) | 10.1-15.8 (3.3-4.5) | 3.7-9.4 (6.5-8.3) | Song et al. (2010) |
| $PM_{2.5}$ | Urban aerosol | ENVI-18, HLB, XAD-8 and DEAE | 37-47 (0.6-1.9) | 33-37 (1.9-3.2) | 13-18 (3.4-4.4) | 5.8-12 (6.5-8.5) | Fan et al. (2014) |
| $PM_{2.5}$ | Urban aerosol | DAX-8 | 43.7-57.5 (0.5-1.9) | 28.3-34.7 (1.9-3.2) | 7.3-15.1 (3.3-4.1) | 0.6-9.1 (6.5-8.3) | Lopes et al. (2015) |

[a] Investigation on the basis of the chemical shift assignments (unit: ppm)

[b] Relative abundance of each type of protons

Table 7. Summary of AAE and MAE$_{365}$ of HULIS and WSOM (sometimes were inferred to water-soluble brown carbon (BrC)).

| Locations | Sample type | HULIS | | WSOM | | References |
|---|---|---|---|---|---|---|
| | | AAE | MAE$_{365}$ (m$^2$/g) | AAE | MAE$_{365}$ (m$^2$/g) | |
| Laboratory | Rice straw smoke PM$_{2.5}$ | 8.2 ±0.6 | 1.54 ±0.30 | 8.1 ±0.8 | 1.24 ±0.33 | |
| Laboratory | Corn straw smoke PM$_{2.5}$ | 6.7 ±0.6 | 2.09 ±0.41 | 6.7 ±0.5 | 1.56 ±0.34 | |
| Laboratory | Pine branch smoke PM$_{2.5}$ | 6.7 ±0.7 | 0.97 ±0.22 | 7.0 ±0.7 | 0.79 ±0.22 | Present work |
| Laboratory | Coal Smoke PM$_{2.5}$ | 13.6 ±0.2 | 0.63 ±0.03 | 13.1 ±0.1 | 0.42 ±0.03 | |
| Guangzhou, China | Urban PM$_{2.5}$ | 7.0 ±0.2 | 0.93 ±0.06 | 6.7 ±0.1 | 0.80 ±0.03 | |
| Laboratory | Rice straw smoke PM$_{2.5}$ | - | - | 8.3 ±0.6 (300-400 nm) | 1.37 ±0.23 | Park and Yu (2016) |
| Laboratory | Pine needles smoke PM$_{2.5}$ | - | - | 7.4 ±1.1 (300-400 nm) | 0.86 ±0.09 | Park and Yu (2016) |
| Laboratory | Sesame stems smoke PM$_{2.5}$ | - | - | 8.0 ±0.8 (300-400 nm) | 1.38 ±0.21 | Park and Yu (2016) |
| Laboratory | Wood smoke particles | - | - | 8.6-17.8 (360-500 nm) | - | Chen et al. (2010) |
| Rondônia, Brazil | BB background PM$_{2.5}$ | 6.4-6.8 (300-700 nm)[a] | - | - | - | Hoffer et al. (2006) |
| Southeastern US | Urban/rural PM$_{2.5}$ | - | - | 6.2-8.3 (330-500 nm) | 0.41-0.87 | Hecobian et al. (2010) |
| Beijing, China | Urban PM$_{2.5}$ | - | - | 7.5±0.9 (330-480 nm) | 1.79±0.24 | Cheng et al. (2011) |
| Beijing, China | Urban PM$_{2.5}$ | - | - | 7.0±0.8 (330-480 nm) | 0.71±0.20 | Cheng et al. (2011) |
| Los Angeles Basin | Urban PM$_{2.5}$ | - | - | 7.58±0.49 (300-600 nm) | 0.70-0.73 | Zhang et al. (2013) |
| Atlanta, USA | Urban/rural PM$_{2.5}$ | - | - | - | 0.13-0.53 | Liu et al. (2013) |
| Beijing, China | Urban PM$_{2.5}$ | - | - | - | 0.51-1.26 | Du et al. (2014) |
| Gosan, Korea | Rural PM$_{2.5}$ and TSP | - | - | 5.6-7.7 (330-400 nm) | 0.3-1.1 | Kirillova et al. (2014) |
| Beijing, China | Urban PM$_{2.5}$ | - | - | 5.83±0.51 (330-400 nm) | 0.73 ±0.15 | Yan et al. (2015) |
| Beijing, China | Urban PM$_{2.5}$ | - | - | 5.30±0.44 (330-400 nm) | 1.54 ±0.16 | Yan et al. (2015) |
| Beijing, China | Urban PM$_{2.5}$ | - | - | 7.28±0.24 (310-450 nm) | 1.22±0.11 | Cheng et al. (2016) |
| Seoul,Korea | Urban PM$_{2.5}$ | - | - | 5.84-9.17 (300-700 nm) | 0.28-1.18 | Kim et al. (2016) |

[a] Representing the range of wavelength chosen for fitting.