# Peer review of "Comprehensive characterization of humic-like substances in smoke PM2.5 emitted from the combustion of biomass materials and fossil fuels"

_Atmospheric Chemistry and Physics, 2016_

## Referee Comment (RC1) · AC Duarte (Referee) · 17 Jun 2016

1. General comments

This comprehensive characterization of humic-like substances (HULIS) in smoke PM2.5 emitted from the combustion of biomass materials and fossil fuels becomes a highly recommendable work for publication since it is really "comprehensive" by using several spectroscopies besides elemental analysis. The authors avoid the discussion WSOC vs HULIS vs. WSOM (water soluble organic matter ) although the filter samples were ultrasonically extracted with Milli-Q water, the water soluble extracts were filtered,

the filtrate was adjusted to a pH value of 2, and finally introduced in a SPE (Oasis HLB) cartridge. When comparing this HULIS isolation procedure with the WSOM isolation procedure suggested by (Duarte and Duarte, 2005) a comment should be made regarding the operational definition of HULIS as different sorbents extract different organic moieties as already shown exactly by the authors of this submitted paper for discussion (Fan et al., 2013).

The variety of materials studied, biomass materials (rice straw, corn straw, and pine branch) and fossil fuels (lignite coal and diesel fuel) allow substantiating the discussion and reach quite accurate and meaningful conclusions regarding the characterization of HULIS from biomass burning (BB) and HULIS from coal combustion. Furthermore, in addition to the smoke PM2.5 samples from the BB and fuel combustion, ambient PM2.5 samples were also collected during December, 2015, which allowed the authors to discuss similarities and differences between primary HULIS and atmospheric HULIS.

Regarding the use of UV-Vis, fluorescence and 1H-NMR spectroscopies for the characterization of properties of HULIS from atmospheric aerosols there are very recent references that also shed some light on this matter and they should be considered when discussing the results of the submitted paper. For the case UV-Vis spectroscopy there is a reference (Matos et al., 2015b) where the authors used comprehensive two-dimensional liquid chromatography (LCxLC) coupled with a diode array detector (DAD) in order to assessing the chemical heterogeneity and mapping the hydrophobicity vs. molecular weight distribution of the most hydrophobic acid fractions in WSOC (which the authors consider as water soluble organic matter, WSOM) from fine atmospheric aerosols collected over different seasons at a urban location. For the case of fluorescence spectroscopy there are two important recent references (Matos et al., 2015a; Paula et al., 2016). In the first reference (Matos et al., 2015a) the authors combine fluorescence datasets of excitation-emission matrices (EEM) fluorescence spectra and Parallel Factor Model (PARAFAC) with Alternating Least Squares (ALS) algorithm in order to further compare sets of excitation-emission matrices fluorescence

spectra of WSOC and Alkaline Soluble Organic Carbon (ASOC), sequentially extracted from urban aerosols collected during different seasons. In the reference (Paula et al., 2016) the authors use a comprehensive multidimensional analysis approach for exploiting simultaneously the compositional changes over a molecular size continuum and associated light-absorption (ultraviolet absorbance and fluorescence) properties of WSOM and alkaline soluble organic matter (ASOM), sequentially extracted from urban aerosols collected during different seasons. For the case of 1H-NMR spectroscopy there are the following two references: a) (Lopes et al., 2015) where the authors applied 1H NMR spectroscopy to characterize the structural features of WSOC and ASOC, sequentially extracted, from fine urban atmospheric aerosols collected over different seasons, and further assess their sources through the pre-established 1H NMR source apportionment fingerprinting approach; and b) (Duarte et al., 2008) where the authors applied 2D NMR techniques to deliver new qualitative information on the substructures present in water soluble organic matter from fine rural atmospheric aerosols. Furthermore the following reference (Duarte and Duarte, 2015) should also be considered as an appropriate background on the application of NMR spectroscopy for acquiring detailed structural characterization of the complex natural organic matter contained in atmospheric aerosols. Finally, the reference (Duarte et al., 2015) should also be taken into account when interpreting the structural features of HULIS from the acquired FTIR spectra (section 3.5). In this reference, the authors used two-dimensional (2D) correlation spectroscopy, applied to one-dimensional solid-state cross polarization magic angle spinning (CP-MAS) 13C NMR, FTIR, and the combination of CP-MAS 13C NMR and FTIR data, to glean new structural information on the most hydrophobic water-soluble organic matter (WSOM) from fine urban air particles collected during different seasons.

All in all, the results add up to an already extensive data set of characteristics of WSOM of aerosols but this work brings an original and comprehensive comparison between excellent proxies for primary HULIS representing biomass burning and fossil fuel combustion and ambient air samples analyzed by several spectroscopies. Therefore the

differences found between primary and atmospheric HULIS as well as the differences found among the four types of primary HULIS can be considered well validated and this study contributes to a better understanding of the differences in chemical nature of primary HULIS from direct combustion emissions and their contribution to atmospheric HULIS.

2. Specific comments

The Abstract should contain more quantitative data and instead of sentences like "HULIS accounted for a significant proportion of the particle matter, . . ." (line 17, page 1) the figures resulting from thus study should be included.

There should be a clarification of the terms HULIS, WSOC and WSOM and not only of HULIS as mentioned in page 2, lines 17 and 18, WSOC as mentioned in page 3, lines 3 and 4. Furthermore, in page 9, lines 21-23, the authors perform an estimate of the contribution of the so-called HULIS to WSOC, using TOC and UV-vis absorbance at 250 nm (UV250) measurements. While the use of TOC is most appropriate for such a comparison, additional details on the use of UV250 measurements should be provided, namely if the UV data were normalized to the amount of carbon of each sample/fraction. This is important for the sake of easier comparison between HULIS and WSOC fractions, mostly because HULIS is an isolated/concentrated fraction of WSOC and it is enriched in those organic moieties preferably retained by the SPE sorbent.

In page 8, line 21 there is the following text "The abundance of HULIS in rice straw smoke PM2.5 (10.7 ± 0.8%). . ." but according to Table 1, this value is for the urban PM2.5 samples and not for rice straw smoke PM2.5. Please correct this inconsistency.

In page 11, lines 18 to 21, the lower OM/OC ratio reported also suggests that primary HULIS in coal smoke are less oxidized than those of HULIS in BB smoke PM2.5. It is difficult to assess the degree of unsaturated components on the basis of the elemental composition data. Besides, no discussion is found regarding the elemental analysis

data (including atomic ratios and OM/OC ratio) of the collected ambient HULIS. For this sample, the values depicted in Table 1 are different from those reported for primary HULIS. A discussion on these differences and values should be included in the manuscript.

The sentence in page 12, lines 2 to 4 ("In contrast, the HULIS in coal smoke had a relatively lower content of O-containing components and a higher content of unsaturated structures.") should be deleted since it is not adding any new valuable information into the discussion.

In page 18, lines 17 to 19: the authors should identify these signals ("sharp peaks") in their 1H NMR spectra in Figure 4. This will be most helpful for readers not familiarized with the interpretation of NMR data. An additional remark on this issue: the identification of single compounds in complex NMR spectra, such as those of Figure 4, is highly arguable. Therefore, the authors should also make this comment in the discussion.

In page 20, paragraph in lines 1 to 5: A more in-depth comparison should be made between NMR data obtained for primary and ambient HULIS.

In page 20, line 10: Please, be aware that this group [H-C-C=] is more likely to have contributions from carbonyl groups (C=O) than from aromatic carbons. Please, consult the original reference (Decesari et al., 2007) for the approach adopted in order to distinguish between these two contributions.

In page 22, paragraph in lines 5 and 6 ("The similarities between 5 primary and atmospheric HULIS suggest they have similar chemical compositions and properties"): at least, this means that they share similar proton functional groups; however, they differ in terms of their relative distribution among the different samples.

3. References

Decesari, S., Mircea, M., Cavalli, F., Fuzzi, S., Moretti, F., Tagliavini, E. and Facchini, M. C.: Source Attribution of Water-Soluble Organic Aerosol by Nuclear

Magnetic Resonance Spectroscopy, Environ. Sci. Technol., 41(7), 2479–2484, doi:10.1021/es061711l, 2007.

Duarte, R. M. B. O. and Duarte, A. C.: Application of Non-Ionic Solid Sorbents (XAD Resins) for the Isolation and Fractionation of Water-Soluble Organic Compounds from Atmospheric Aerosols, J. Atmos. Chem., 51(1), 79–93, doi:10.1007/s10874-005-8091-x, 2005.

Duarte, R. M. B. O. and Duarte, A. C.: Unraveling the structural features of organic aerosols by NMR spectroscopy: a review, Magn. Reson. Chem., 53(9), 658–666, doi:10.1002/mrc.4227, 2015.

Duarte, R. M. B. O., Silva, A. M. S. and Duarte, A. C.: Two-Dimensional NMR Studies of Water-Soluble Organic Matter in Atmospheric Aerosols, Environ. Sci. Technol., 42(22), 8224–8230, doi:10.1021/es801298s, 2008.

Duarte, R. M. B. O., Freire, S. M. S. C. and Duarte, A. C.: Investigating the water-soluble organic functionality of urban aerosols using two-dimensional correlation of solid-state 13C NMR and FTIR spectral data, Atmos. Environ., 116, 245–252, doi:10.1016/j.atmosenv.2015.06.043, 2015.

Fan, X., Song, J. and Peng, P.: Comparative study for separation of atmospheric humic-like substance (HULIS) by ENVI-18, HLB, XAD-8 and DEAE sorbents: Elemental composition, FT-IR, 1H NMR and off-line thermochemolysis with tetramethylammonium hydroxide (TMAH), Chemosphere, 93(9), 1710–1719, doi:10.1016/j.chemosphere.2013.05.045, 2013.

Lopes, S. P., Matos, J. T. V., Silva, A. M. S., Duarte, A. C. and Duarte, R. M. B. O.: 1H NMR studies of water- and alkaline-soluble organic matter from fine urban atmospheric aerosols, Atmos. Environ., 119, 374–380, doi:10.1016/j.atmosenv.2015.08.072, 2015.

Matos, J. T. V., Freire, S. M. S. C., Duarte, R. M. B. O. and Duarte, A. C.: Natural organic matter in urban aerosols: Comparison between water and alkaline soluble components

using excitation–emission matrix fluorescence spectroscopy and multiway data analysis, Atmos. Environ., 102, 1–10, doi:10.1016/j.atmosenv.2014.11.042, 2015a.

Matos, J. T. V., Freire, S. M. S. C., Duarte, R. M. B. O. and Duarte, A. C.: Profiling Water-Soluble Organic Matter from Urban Aerosols Using Comprehensive Two-Dimensional Liquid Chromatography, Aerosol Sci. Technol., 49(6), 381–389, doi:10.1080/02786826.2015.1036394, 2015b.

Paula, A. S., Matos, J. T. V., Duarte, R. M. B. O. and Duarte, A. C.: Two chemically distinct light-absorbing pools of urban organic aerosols: A comprehensive multidimensional analysis of trends, Chemosphere, 145, 215–223, doi:10.1016/j.chemosphere.2015.11.093, 2016.

---

## Referee Comment (RC2) · Anonymous Referee #2 · 20 Jun 2016

General Comments: This study discusses comprehensive characterization of humic-like substances (HULIS) in PM2.5 samples from combustions of biomass materials (rice straw, corn straw, and pine branch) and fossil fuels (lignite coal and diesel fuel), and from ambient air. To achieve the goals of this study, water-soluble HULIS fractions were group isolated using a HLB solid phase extraction method and then quantified with a TOC analyzer. Also chemical properties and structures of HULIS were further investigated using elemental analyzer, UV-vis spectroscopy, excitation-emissions matrix (EEM) fluorescence spectroscopy, FTIR spectroscopy, and 1H-NMR spectroscopy. Characteristics of primary HULIS from biomass burning (BB) and fossil fuel (FF) com-

bustion emissions were compared with the result from ambient samples and with those reported from many previous publications. Results indicate the chemical properties and structures of primary HULIS from combustion emissions of biomass and fossil fuels are very similar to chemical features of ambient HULIS in this and previous studies, which are indicated by a variety of analytical tools, with some distinct differences. It is worthy of note that primary HULIS contain mostly low molecular weight compounds. Results from this study can add to the database of chemical properties and structures for BB and FF-derived HULIS, and thereby contribute to better understanding of the role of BB and FF aerosols in ambient environments. Also this work may help to identify future focus in related to molecular level characterization of ambient brown carbon. However, most of the findings from this study were demonstrated by previous publications. Thus, authors should address the unique scientific finding of this work a bit more in revised manuscript. Overall the manuscript is written well, and with some further explanation of collected data and further elaboration on the results it will be ready for publication. Below are specific revision comments for the authors to consider in their next revision.

Specific comments: Abstract section I would suggest adding important quantitative information from the study.

1. Introduction section Park and Yu (2016) examined the chemical and light absorption properties of HULIS in PM2.5 from burning of three different types of biomass burning fuels (rice straw, pine needles, and sesame branch) in a laboratory combustion chamber ("Chemical and light absorption properties of humic-like substances from biomass burning emissions under controlled combustion experiments". Atmospheric Environment 136, 114-122). Authors may refer to the paper to compare their results.

2. Experimental section 2.1. Sampling (pages 4-5) Lines 11-12 on page 4: It is described that "..five types of smoke PM2.5 samples were collected to . . .from the combustion of biomass .." How many sets of PM2.5 samples did you collect for each of biomass types and coal fuel? Need to be added in the revision. Was only one sample used for each burn to investigate the comprehensive characterization of HULIS in smoke aerosols samples? If so, they should describe the reliability and/or uncertainties of the experimental results. Also how many ambient samples did you use to conduct the experiments?

Combustion conditions of three biomass materials and lignite coal in a laboratory re-suspension chamber should be described in detail because the burning conditions such as smoldering or flaming burns, combustion temperature, air dilution ratio, flue gas temperature at a sampling location, etc., affect greatly the abundance and chemical properties and structures of WSOC, HULIS, and organic compounds. Also burning conditions might generate water-soluble aerosols of different optical properties. Details in this regard would be helpful.

At what stage of the burning were the samples collected? Please be as specific as possible.

What were the moisture contents of the biomass burning and coal fuels? The authors need to describe the elemental composition (C, H, O, N, and S) of burning fuels if possible, but for comparison with other papers moisture content would be very helpful.

Please add collection time for biomass smokes.

2.2. Isolation of HULIS Lines 5-6 on page 6: "...more filters were used to obtain HULIS for the analysis of the elemental composition..." Instead of using the HULIS samples re-dissolved in 20 mL Milli-Q water (section 2.2), new filter samples were used for further analyses? More detailed description would be helpful for readers.

2.3.2. Elemental composition Lines 20-21 on page 6: "A portion of the HULIS (re-dissolved in methanol) was transferred into ...." The HULIS eluate used in this analysis was not re-dissolved in water? How much volume of MeOH did you use for this?

2.3.6. 1H-NMR spectroscopy "About 10 mg of dried HULIS were re-dissolved in 1 mL of MeOD." could be changed to "....of MeOH."

3. Results and discussion 3.1. The abundance of HULIS in smoke PM2.5 and ambient PM2.5 These results should be compared with those from Park and Yu (2016). In Table 1, please include number of samples used in the experiments for each of BB, FF, and ambient samples. References of Park et al. in Table 1 are not listed in the list of the references.

3.2. Elemental composition Lines 15-21 on page 11: In table 2, OM/OC ratios for four types of primary HULIS are presented. They did not measure OC concentration. Details how OM/OC ratios got determined from elemental composition data should described in the text.

3.3 UV-vis properties & 3.4 Fluorescence properties I think that authors measured light absorption spectra of WSOC and HULIS from BB, FF, and ambient samples. I would suggest providing absorption angstrom exponents (AAE) and mass absorption efficiencies (MAE) of samples from burning of different types of biomass and coal fuels, and ambient environment. These information could be much useful for understanding light absorption characteristics and radiative forcing effects by BB and coal burning-derived brown carbon aerosols.

Lines 19-20 on page 13 and lines 10-11 on page 15. Authors stated that based on the SUVA254 values from primary smoke HULUS, "the primary HULIS contained higher aromatic degree and/or higher MW compounds", but results from EEM spectra indicate that "primary HULIS contain mostly low MW compounds". This means that primary HULIS from BB and FF smokes contain both high and low MW compounds? Further elaboration on this is needed.

3.7 Comparison pf primary HULIS and 4 Conclusions Sections 3.7.1, 3.7.2, and 3.7.3 are very similar to the explanations in sections 3.2-3.6, so it needs to be condensed, or I suggest combining the section 3.7 with section 4. Conclusions.

4. Conclusions It will be much more valuable if a paragraph was added to conclusions describing what the authors think was important and how it can be applied.

---

## Author Comment (AC1) · 24 Aug 2016

Dear AC Duarte,

Thanks for your interests and helpful comments in our paper. We have carefully revised the manuscript according to the comments. A point-to-point response to the comments is given below. We hope we answered your questions adequately.

[Figure]

1. General comments

This comprehensive characterization of humic-like substances (HULIS) in smoke PM2.5 emitted from the combustion of biomass materials and fossil fuels becomes a highly recommendable work for publication since it is really "comprehensive" by using several spectroscopies besides elemental analysis. The authors avoid the discussion WSOC vs HULIS vs. WSOM (water soluble organic matter ) although the filter samples were ultrasonically extracted with Milli-Q water, the water soluble extracts were filtered, the filtrate was adjusted to a pH value of 2, and finally introduced in a SPE (Oasis HLB) cartridge. When comparing this HULIS isolation procedure with the WSOM isolation procedure suggested by (Duarte and Duarte, 2005) a comment should be made regarding the operational definition of HULIS as different sorbents extract different organic moieties as already shown exactly by the authors of this submitted paper for discussion (Fan et al., 2013).

Reply: Thanks for the comments. We have added some operational definitions and comments when comparing this HULIS isolation procedure with the WSOM isolation procedure suggested by (Duarte and Duarte, 2005) and others in revised manuscript. In addition, the HULIS isolation procedures have been added in the Table 2, 3 and 4 of the revised manuscript. The revisions stated in our revised manuscript are as below:

Page 3, lines 9-10: "Their carbon (C) mass accounts for 9%–72% of the C content of water soluble organic matter (WSOM) in atmospheric aerosols (Feczko et al., 2007; Krivacsy et al., 2008; Lin et al., 2010b; Fan et al., 2012; Song et al., 2012)."

Page 3, line 20- Page 4, line 1: "HULIS fractions have been found in smoke particles emitted from the combustion of wood and leaves, the carbon content of HULIS (HULIS-C) make up 0.6–21.2% of the total mass of particles (Schmidl et al., 2008a; Schmidl et al., 2008b; Goncalves et al., 2010). HULIS accounted for 7.6–12.4% of the particle mass, and HULIS-C contributed approximate 14.3–14.7% and 30–33% of the organic carbon (OC) and the water soluble carbon (WSOC), respectively."

Moreover, a comment regarding the operational definition of HULIS has been added. The revisions are as follow:

Page 7, lines 19- Page 8, lines 1: "It is noted that the eluates here represent the hydrophobic portion of WSOM and were named as water soluble HULIS. According to the literatures (Graber and Rudich, 2006; Zheng et al., 2013), these water soluble hydrophobic WSOM can be isolated with different SPE methods. In spite of some differences were observed among of them, these hydrophobic WSOM isolated with different sorbents are very similar in chemical compositions and properties according to our previous studies (Fan et al., 2012, 2013). Therefore, for better comparison with other studies, the hydrophobic WSOM isolated by SPE methods (i.e. HLB, C-18, DEAE, XAD-8) and other protocols (i.e. ELSD) are all termed as HULIS in this paper."

The variety of materials studied, biomass materials (rice straw, corn straw, and pine branch) and fossil fuels (lignite coal and diesel fuel) allow substantiating the discussion and reach quite accurate and meaningful conclusions regarding the characterization of HULIS from biomass burning (BB) and HULIS from coal combustion. Furthermore, in addition to the smoke PM2.5 samples from the BB and fuel combustion, ambient PM2.5 samples were also collected during December, 2015, which allowed the authors to discuss similarities and differences between primary HULIS and atmospheric HULIS. Regarding the use of UV-Vis, fluorescence and 1H-NMR spectroscopies for the characterization of properties of HULIS from atmospheric aerosols there are very recent references that also shed some light on this matter and they should be considered when discussing the results of the submitted paper. For the case UV-Vis spectroscopy there is a reference (Matos et al., 2015b) where the authors used comprehensive two-dimensional liquid chromatography (LCxLC) coupled with a diode array detector (DAD) in order to assessing the chemical heterogeneity and mapping the hydrophobicity vs. molecular weight distribution of the most hydrophobic acid fractions in WSOC (which the authors consider as water soluble organic matter, WSOM) from fine atmospheric aerosols collected over different seasons at a urban location. For the case

of fluorescence spectroscopy there are two important recent references (Matos et al., 2015a; Paula et al., 2016). In the first reference (Matos et al., 2015a) the authors combine fluorescence datasets of excitation-emission matrices (EEM) fluorescence spectra and Parallel Factor Model (PARAFAC) with Alternating Least Squares (ALS) algorithm in order to further compare sets of excitation-emission matrices fluorescence spectra of WSOC and Alkaline Soluble Organic Carbon (ASOC), sequentially extracted from urban aerosols collected during different seasons. In the reference (Paula et al., 2016) the authors use a comprehensive multidimensional analysis approach for exploiting simultaneously the compositional changes over a molecular size continuum and associated light-absorption (ultraviolet absorbance and fluorescence) properties of WSOM and alkaline soluble organic matter (ASOM), sequentially extracted from urban aerosols collected during different seasons. For the case of 1H-NMR spectroscopy there are the following two references: a) (Lopes et al., 2015) where the authors applied 1H NMR spectroscopy to characterize the structural features of WSOC and ASOC, sequentially extracted, from fine urban atmospheric aerosols collected over different seasons, and further assess their sources through the pre-established 1H NMR source apportionment fingerprinting approach; and b) (Duarte et al., 2008) where the authors applied 2D NMR techniques to deliver new qualitative information on the substructures present in water soluble organic matter from fine rural atmospheric aerosols. Furthermore the following reference (Duarte and Duarte, 2015) should also be considered as an appropriate background on the application of NMR spectroscopy for acquiring detailed structural characterization of the complex natural organic matter contained in atmospheric aerosols.

Reply: Thanks a lot. We have revised the text according to the comments and added some recent references when discussing the results in revised manuscript. The revisions stated in our revised manuscript are as below:

Page 16, lines 22- Page 17, lines 2: "These results indicate that the primary HULIS contain more aromatic groups with conjugation of $\pi$-bonds alongside aliphatic structures. Similar characters were also found in many previous studies. For example, it has been found that HULIS fraction in colder season presented more aromatic structures than those in warmer season, of which the BB might be an important contribution of the former one (Baduel et al., 2010; Matos et al., 2015a, b; Paula et al., 2016)."

Page 20, lines 7-9: "The bands at 1458 and 1610 cm-1, which are generally attributed to the C–C stretching of aromatic rings (Watanabe and Kuwatsuka, 1992; Duarte et al., 2015), are observed in spectra of primary HULIS from direct combustion emissions."

Page 21, lines 15-18: "A critical review on the application of 1H NMR spectroscopy on WSOM in atmospheric aerosols has been presented, in which 1H NMR was demonstrated to be an important and novel tool to characterize WSOM, which can not only provide deeply insight into the structural characteristics of them but also reflect their sources (Duarte and Duarte, 2015)."

Page 21, lines 22- Page 22, lines 1: "The four types of primary HULIS displayed similar spectra to the atmospheric HULIS spectra in this study, which were also comparable to 1H NMR spectra of HULIS and/or WSOM in fog (Decesari et al., 2000), cloud (Decesari et al., 2005), rain water (Santos et al., 2009, 2012), biomass burning aerosols (Graham et al., 2002) and urban/rural aerosol (Decesari et al., 2007; Ziemba et al., 2011;Song et al., 2012; Fan et al., 2013; Lopes et al., 2015)."

Page 22, lines 9-11: "Among these sharp signals, a limited number of resonances could be attributed to specific organic species by comparison with previous studies (Decesari et al., 2000, 2001; Suzuki et al., 2001; Matta et al., 2003; Cavalli et al., 2006; Chalbot et al., 2014, 2016ïijŻLopes et al., 2015)"

Page 22, line 24-Page 23, line 4: "This suggests that HULIS consists of a complex mixture of organic substances (Samburova et al., 2007; Song et al., 2012; Fan et al., 2013; Lopes et al., 2015). The integrated 1H NMR signal over specific ranges of chemical shift has been used previously to quantify the contribution of organic functional groups in HULIS from urban/rural aerosols (Song et al., 2012; Fan et al., 2013; Lopes et al.,

2015) and rainwater (Miller et al., 2009; Santos et al., 2009, 2012)."

Page 23, lines 18-24: "These four functional groups have also been observed in the 1H NMR spectra of atmospheric HULIS in this and other studies, but the relative distribution of these four functional groups are different. Whether atmospheric HULIS in this work or in other studies from ambient aerosol (Song et al., 2012; Fan et al., 2013; Lopes et al., 2015) and rainwater (Miller et al., 2009; Santos et al., 2009, 2012) were all characterized by a predominance of [R-H] (37–60%), followed by [H–C–C=] (20–37%) and [H–C–O] (10–24%), and a less contribution from [Ar–H] (1–12%)."

Page 24, Line 2-4: "This result is also consistent with the observations of more aromatic protons in HULIS in colder seasons ascribed to BB influence (Song et al., 2012; Lopes et al., 2015)."

We also added more information (red) in Table 6 in revised manuscript.

Some new references have been added in revised manuscript:

Duarte, R. M., and Duarte, A. C.: Unraveling the structural features of organic aerosols by NMR spectroscopy: a review, Magn Reson Chem, 53, 658-666, 10.1002/mrc.4227, 2015.

Duarte, R. M. B. O., Freire, S. M. S. C., and Duarte, A. C.: Investigating the water-soluble organic functionality of urban aerosols using two-dimensional correlation of solid-state 13C NMR and FTIR spectral data, Atmos. Environ., 116, 245-252, 10.1016/j.atmosenv.2015.06.043, 2015.

Lopes, S. P., Matos, J. T. V., Silva, A. M. S., Duarte, A. C., and Duarte, R. M. B. O.: 1H NMR studies of water- and alkaline-soluble organic matter from fine urban atmospheric aerosols, Atmos. Environ., 119, 374-380, 10.1016/j.atmosenv.2015.08.072, 2015.

Matos, J. T. V., Freire, S. M. S. C., Duarte, R. M. B. O., and Duarte, A. C.: Natural organic matter in urban aerosols: Comparison between water and alkaline soluble components using excitation–emission matrix fluorescence spectroscopy and multiway data analysis, Atmos. Environ., 102, 1-10, http://dx.doi.org/10.1016/j.atmosenv.2014.11.042, 2015a.

Matos, J. T. V., Freire, S. M. S. C., Duarte, R. M. B. O., and Duarte, A. C.: Profiling Water-Soluble Organic Matter from Urban Aerosols Using Comprehensive Two-Dimensional Liquid Chromatography, Aerosol Sci. Technol., 49, 381-389, 10.1080/02786826.2015.1036394, 2015b.

Paula, A. S., Matos, J. T., Duarte, R. M., and Duarte, A. C.: Two chemically distinct light-absorbing pools of urban organic aerosols: A comprehensive multidimensional analysis of trends, Chemosphere, 145, 215-223, 10.1016/j.chemosphere.2015.11.093, 2016.

Finally, the reference (Duarte et al., 2015) should also be taken into account when interpreting the structural features of HULIS from the acquired FTIR spectra (section 3.5). In this reference, the authors used two-dimensional (2D) correlation spectroscopy, applied to one-dimensional solid-state cross polarization magic angle spinning (CP-MAS) 13C NMR, FTIR, and the combination of CP-MAS 13C NMR and FTIR data, to glean new structural information on the most hydrophobic water soluble organic matter (WSOM) from fine urban air particles collected during different seasons.

Reply: Thanks. We have revised the text when interpreting the structural features of HULIS from the acquired FTIR spectra and added this reference in revised manuscript. The details are as below:

Page 19, lines 11-13: "As shown in Figure 3, the spectra of primary HULIS were similar to those of the atmospheric HULIS and WSOM in this and previous studies (Havers et al., 1998; Krivacsy et al., 2001; Kiss et al., 2002; Duarte et al., 2005; Polidori et al., 2008; Santos et al., 2009; Fan et al., 2013; Duarte et al., 2015)."

Page 20, line 1-3: "These FTIR spectra features are similar to those of the atmospheric HULIS described in this and other studies (Krivacsy et al., 2001; Duarte et al., 2005;

Song et al., 2012; Fan et al., 2013; Duarte et al., 2015)."

Page 20, lines 5-9: "Relative weaker band at 1710 cm-1 for primary HULIS than for atmospheric HULIS was observed in Figure 3, indicating the former ones present less carboxyl groups (Song et al., 2012; Fan et al., 2013). The bands at 1458 and 1610 cm-1, which are generally attributed to the C–C stretching of aromatic rings (Watanabe and Kuwatsuka, 1992; Duarte et al., 2015), are observed in spectra of primary HULIS from direct combustion emissions."

The reference has been added in revised manuscript:

Duarte, R. M. B. O., Freire, S. M. S. C., and Duarte, A. C.: Investigating the water-soluble organic functionality of urban aerosols using two-dimensional correlation of solid-state 13C NMR and FTIR spectral data, Atmos. Environ., 116, 245-252, 10.1016/j.atmosenv.2015.06.043, 2015.

All in all, the results add up to an already extensive data set of characteristics of WSOM of aerosols but this work brings an original and comprehensive comparison between excellent proxies for primary HULIS representing biomass burning and fossil fuel combustion and ambient air samples analyzed by several spectroscopies. Therefore the differences found between primary and atmospheric HULIS as well as the differences found among the four types of primary HULIS can be considered well validated and this study contributes to a better understanding of the differences in chemical nature of primary HULIS from direct combustion emissions and their contribution to atmospheric HULIS.

2. Specific comments

The Abstract should contain more quantitative data and instead of sentences like "HULIS accounted for a significant proportion of the particle matter, . . ." (line 17, page 1) the figures resulting from thus study should be included.

Reply: Thanks. We have revised the abstract according to the comments and more

quantitative data were included in the abstract.

Page 1, lines 17-20: "The results show that HULIS account for 11.2–23.7% and 5.3% of PM2.5 emitted from biomass burning (BB) and coal combustion, respectively. In addition, contributions of HULIS-C to total carbon and water soluble carbon in smoke PM2.5 emitted from BB and coal combustion are 8.0–21.7% and 5.2%, 56.9–66.1% and 45.5%, respectively."

There should be a clarification of the terms HULIS, WSOC and WSOM and not only of HULIS as mentioned in page 2, lines 17 and 18, WSOC as mentioned in page 3, lines 3 and 4.

Reply: Thanks for the comments. We have added the definitions of HULIS, WSOC and WSOM and made a clarification of the terms HULIS, WSOC and WSOM in revised manuscript. The details are as follows:

Page 3, lines 9-11: "Their carbon (C) mass accounts for 9%–72% of the C content of water soluble organic matter (WSOM) in atmospheric aerosols (Feczko et al., 2007; Krivacsy et al., 2008; Lin et al., 2010b; Fan et al., 2012; Song et al., 2012)"

Page 3, lines 20-22: "HULIS fractions have been found in smoke particles emitted from the combustion of wood and leaves, the carbon content of HULIS (HULIS-C) make up 0.6–21.2% of the total mass of particles (Schmidl et al., 2008a; Schmidl et al., 2008b; Goncalves et al., 2010)"

Page 3, line 22-Page 4, line 1: "HULIS have also been found to be abundant in fresh burning emissions from rice straw and sugar cane leaves (Lin et al., 2010a; Lin et al., 2010b). HULIS accounted for 7.6–12.4% of the particle mass, and HULIS-C contributed approximate 14.3–14.7% and 30–33% of the organic carbon (OC) and the water soluble carbon (WSOC), respectively."

Page 7, line 19- Page 8, line 1: "It is noted that the eluates here represent the hydrophobic portion of WSOM and were named as water soluble HULIS. According to

the literatures (Graber and Rudich, 2006; Zheng et al., 2013), these water soluble hydrophobic WSOM can be isolated with different SPE methods. In spite of some differences were observed among of them, these hydrophobic WSOM isolated with different sorbents are very similar in chemical compositions and properties according to our previous studies (Fan et al., 2012, 2013). Therefore, for better comparison with other studies, the hydrophobic WSOM isolated by SPE methods (i.e. HLB, C-18, DEAE, XAD-8) and other protocols (i.e. ELSD) are all termed as HULIS in this paper."

Moreover, the HULIS isolation procedures have been also added in the revised Table 2, 3 and 4.

Furthermore, in page 9, lines 21-23, the authors perform an estimate of the contribution of the so-called HULIS to WSOC, using TOC and UV-vis absorbance at 250 nm (UV250) measurements. While the use of TOC is most appropriate for such a comparison, additional details on the use of UV250 measurements should be provided, namely if the UV data were normalized to the amount of carbon of each sample/fraction. This is important for the sake of easier comparison between HULIS and WSOC fractions, mostly because HULIS is an isolated/concentrated fraction of WSOC and it is enriched in those organic moieties preferably retained by the SPE sorbent.

Reply: Thanks for the comments. We have added some details on the use of UV250 measurements in the revised manuscript. In the study, the UV250 measurement was used to investigate the contribution of HULIS fraction in total WSOM. The UV data weren't normalized to the amount of carbon of each sample/fraction. The details are as below:

Page 9, lines 9-15: "As demonstrated by many studies, the high absorbing UV chromophoric compounds (strong absorbing at 250 nm) are major components in WSOM, which usually tended to be enriched in the SPE isolated HULIS fractions (Baduel et al., 2009; Fan et al., 2012, 2013, 2016; Song et al., 2012; Duarte et al., 2015; Lopes et al., 2015; Fan et al., 2016). Therefore, the ratio between the UV250 of HULIS and original

WSOM has been widely used to evaluate the relative contribution of HULIS to WSOM in terms of chromophoric compounds content. It should be noted that HULIS solution must keep the same volume to original WSOM solution for the UV250 determination."

In page 8, line 21 there is the following text "The abundance of HULIS in rice straw smoke PM2.5 (10.7 $\pm$ 0.8%). . ." but according to Table 1, this value is for the urban PM2.5 samples and not for rice straw smoke PM2.5. Please correct this inconsistency.

Reply: I am sorry for this mistake. The sentence has been revised as follow:

Page 11, lines 23-25: "The abundance of HULIS in rice straw smoke PM2.5 (23.4 $\pm$ 5.5%) was also significantly higher than in ambient PM2.5 in this study, and in some previous studies (as listed in Table 2)."

In page 11, lines 18 to 21, the lower OM/OC ratio reported also suggests that primary HULIS in coal smoke are less oxidized than those of HULIS in BB smoke PM2.5. It is difficult to assess the degree of unsaturated components on the basis of the elemental composition data.

Reply: Thanks. We have revised the statement and revised the sentences in revised manuscript.

Page 15, lines 2-5: "The HULIS samples in the three types of biomass smoke PM2.5 had a relatively lower C content (52.9–57.4%), higher O content (33.0–38.3%), higher O/C molar ratio (0.43–0.54) and higher OM/OC (1.74–1.89) than those in coal smoke PM2.5. These results indicated that the HULIS in BB smoke contained a relatively higher content of O-containing components."

Besides, no discussion is found regarding the elemental analysis data (including atomic ratios and OM/OC ratio) of the collected ambient HULIS. For this sample, the values depicted in Table 1 are different from those reported for primary HULIS. A discussion on these differences and values should be included in the manuscript.

Reply: Thanks. We have added some discussions for the elemental analysis data of

the collected ambient HULIS in revised manuscript.

Page 13, lines 22-25: "It is obvious that the primary HULIS contain substantially higher C and lower O than ambient HULIS in this study. However, these results are comparable with those for atmospheric HULIS in previous studies (Krivacsy et al., 2001; Kiss et al., 2002; Duarte et al., 2007; Salma et al., 2007; Song et al., 2012; Fan et al., 2013; Duarte et al., 2015)."

Page 14, lines 8-12: "Among the four types of primary HULIS, the O/C ratios of the three types from BB were in the range of 0.43–0.54, which were lower than 0.65 for ambient HULIS in this study, but were comparable with data (0.30–0.76) for atmospheric HULIS reported in previous studies (Krivacsy et al., 2001; Kiss et al., 2002; Duarte et al., 2007; Salma et al., 2007; Song et al., 2012; Fan et al., 2013)."

Page 14, lines 15-24: "The H/C molar ratios of the four primary types of HULIS were in the ranges of 1.15 to 1.43, which were lower than that (1.59) for atmospheric HULIS in this work. However, they dropped in the range of observations (1.01–1.53) reported in previous studies (Krivacsy et al., 2001; Kiss et al., 2002; Duarte et al., 2007, 2015; Salma et al., 2007; Song et al., 2012; Fan et al., 2013). The N/C molar ratios of primary HULIS were 0.03–0.06, with both being similar to the results for atmospheric HULIS in this and previous studies (Table 3). In addition, the ratio of OM to organic C (OM/OC) mass ratios of the four types of primary HULIS ranged from 1.51 to 1.89, which were lower than 2.06 for ambient HULIS in this study, but were generally in the range of the data (1.5–2.28) reported for atmospheric HULIS in previous studies (Krivacsy et al., 2001; Kiss et al., 2002; Duarte et al., 2007; Salma et al., 2007; Song et al., 2012; Fan et al., 2013; Duarte et al., 2015)."

The sentence in page 12, lines 2 to 4 ("In contrast, the HULIS in coal smoke had a relatively lower content of O-containing components and a higher content of unsaturated structures.") should be deleted since it is not adding any new valuable information into the discussion.

Reply: Thanks. We have deleted it.

In page 18, lines 17 to 19: the authors should identify these signals ("sharp peaks") in their 1H NMR spectra in Figure 4. This will be most helpful for readers not familiarized with the interpretation of NMR data. An additional remark on this issue: the identification of single compounds in complex NMR spectra, such as those of Figure 4, is highly arguable. Therefore, the authors should also make this comment in the discussion.

Reply: Thanks. We have added some comments for these signals in the discussion. The paragraph has been revised as follow:

Page 22, lines 11-22: "These sharp signals included low molecular weight formate (8.44 ppm), terephthalic acid (8.01 and 8.12 ppm), phthalic acid (7.45–7.47 and 7.58 ppm), glucose (3.88–3.91 and 3.81–3.85 ppm), fructose (3.79–3.84 ppm), trimethylamine (2.71 and 2.89 ppm), dimethylamine (2.72 ppm), monomethylamine (2.55 ppm). It is worth noting that all BB-derived HULIS present more sharp glucose and fructose resonances than atmospheric HULIS in 1H NMR spectra, but they were absent for coal combustion derived HULIS. On the other hand, coal combustion derived HULIS contain more sharp resonances of terephthalic acid and phthalic acid than atmospheric HULIS, but they were absent for BB-derived HULIS. Moreover, whether BB-derived or coal combustion derived HULIS exhibit many sharp signals between 6.5–8.5 ppm, which could ascribed to aromatic structures, such as substituted phenols and alkylbenzenes (around 6.6–7.0 ppm), benzoic acids, esters, and nitroaromatics (Suzuki et al., 2001; Chalbot et al., 2014)."

We have also revised the Figure 4, and given some identi́cations for some sharp peaks in the revised Figure 4 in current manuscript.

In page 20, paragraph in lines 1 to 5: A more in-depth comparison should be made between NMR data obtained for primary and ambient HULIS.

Reply: Thanks. We have made a more in-depth comparison between NMR data obtained for primary and ambient HULIS in revised manuscript:

Page 23, line 18-page 24, line 4: "These four functional groups have also been observed in the 1H NMR spectra of atmospheric HULIS in this and other studies, but the relative distribution of these four functional groups are different. Whether atmospheric HULIS in this work or in other studies from ambient aerosol (Song et al., 2012; Fan et al., 2013; Lopes et al., 2015) and rainwater (Miller et al., 2009; Santos et al., 2009, 2012) were all characterized by a predominance of [H–C] (37–60%), followed by [H–C–C=] (20–37%) and [H–C–O] (10–24%), and a less contribution from [Ar–H] (1–12%). It was noted that the relative content of [Ar–H] groups (19–31%) in primary HULIS was significantly higher than that in atmospheric HULIS. This suggests that primary HULIS contained more aromatic structures, which is consistent with the elemental composition, UV-vis spectra, EEM fluorescence spectra, and FTIR spectra results. This result is also consistent with the observations of more aromatic protons in HULIS in colder seasons ascribed to BB influence (Song et al., 2012; Lopes et al., 2015)."

In page 20, line 10: Please, be aware that this group [H-C-C=] is more likely to have contributions from carbonyl groups (C=O) than from aromatic carbons. Please, consult the original reference (Decesari et al., 2007) for the approach adopted in order to distinguish between these two contributions.

Reply: Thanks. We have revised that in current manuscript.

Page 24, lines 6-9: "As shown in Table 6, there was a relatively higher content of [H–C–C=] (40%) and [Ar–H] (31%) in primary HULIS from coal combustion than from BB, indicating that the primary HULIS in coal smoke contained more unsaturated aliphatic (i.e., carbonyl groups (C=O)) and aromatic structural groups."

Page 24, line 16-line 20: "For the HULIS from BB, a relatively high content of [C–H] and low content of [Ar–H] were observed for HULIS from pine branch combustion when compared to the HULIS from rice straw and corn straw combustion. These results suggest that the primary HULIS from pine branch combustion contained relative higher

content of aliphatic protons and lower content of aromatic protons than the HULIS from rice straw and corn straw combustion."

In page 22, paragraph in lines 5 and 6 ("The similarities between 5 primary and atmospheric HULIS suggest they have similar chemical compositions and properties"): at least, this means that they share similar proton functional groups; however, they differ in terms of their relative distribution among the different samples.

Reply: Thanks. This is an inexact conclusion. So we have deleted this sentence in revised manuscript.

3. References

Decesari, S., Mircea, M., Cavalli, F., Fuzzi, S., Moretti, F., Tagliavini, E. and Facchini, M. C.: Source Attribution of Water-Soluble Organic Aerosol by Nuclear Magnetic Resonance Spectroscopy, Environ. Sci. Technol., 41(7), 2479–2484, doi:10.1021/es061711l, 2007.

Duarte, R. M. B. O. and Duarte, A. C.: Application of Non-Ionic Solid Sorbents (XAD Resins) for the Isolation and Fractionation of Water-Soluble Organic Compounds from Atmospheric Aerosols, J. Atmos. Chem., 51(1), 79–93, doi:10.1007/s10874-005-8091-x, 2005.

Duarte, R. M. B. O. and Duarte, A. C.: Unraveling the structural features of organic aerosols by NMR spectroscopy: a review, Magn. Reson. Chem., 53(9), 658–666, doi:10.1002/mrc.4227, 2015.

Duarte, R. M. B. O., Silva, A. M. S. and Duarte, A. C.: Two-Dimensional NMR Studies of Water-Soluble Organic Matter in Atmospheric Aerosols, Environ. Sci. Technol., 42(22), 8224–8230, doi:10.1021/es801298s, 2008.

Duarte, R. M. B. O., Freire, S. M. S. C. and Duarte, A. C.: Investigating the watersoluble organic functionality of urban aerosols using two-dimensional correlation of solid-state 13C NMR and FTIR spectral data, Atmos. Environ., 116, 245–252,

doi:10.1016/j.atmosenv.2015.06.043, 2015.

Fan, X., Song, J. and Peng, P.: Comparative study for separation of atmospheric humic-like substance (HULIS) by ENVI-18, HLB, XAD-8 and DEAE sorbents: Elemental composition, FT-IR, 1H NMR and off-line thermochemolysis with tetramethylammonium hydroxide (TMAH), Chemosphere, 93(9), 1710–1719, doi:10.1016/j.chemosphere.2013.05.045, 2013.

Lopes, S. P., Matos, J. T. V., Silva, A. M. S., Duarte, A. C. and Duarte, R. M. B. O.: 1H NMR studies of water- and alkaline-soluble organic matter from fine urban atmospheric aerosols, Atmos. Environ., 119, 374–380, doi:10.1016/j.atmosenv.2015.08.072, 2015.

Matos, J. T. V., Freire, S. M. S. C., Duarte, R. M. B. O. and Duarte, A. C.: Natural organic matter in urban aerosols: Comparison between water and alkaline soluble components using excitation–emission matrix fluorescence spectroscopy and multiway data analysis, Atmos. Environ., 102, 1–10, doi:10.1016/j.atmosenv.2014.11.042, 2015a.

Matos, J. T. V., Freire, S. M. S. C., Duarte, R. M. B. O. and Duarte, A. C.: Profiling Water-Soluble Organic Matter from Urban Aerosols Using Comprehensive Two-Dimensional Liquid Chromatography, Aerosol Sci. Technol., 49(6), 381–389, doi:10.1080/02786826.2015.1036394, 2015b.

Paula, A. S., Matos, J. T. V., Duarte, R. M. B. O. and Duarte, A. C.: Two chemically distinct light-absorbing pools of urban organic aerosols: A comprehensive multidimensional analysis of trends, Chemosphere, 145, 215–223, doi:10.1016/j.chemosphere.2015.11.093, 2016.

---

## Author Comment (AC2) · 24 Aug 2016

We are grateful to Anonymous Referee #2 for his/her valuable comments, and have carefully revised our manuscript accordingly. A point-to-point response to this reviewer's comments is given below.

General Comments: This study discusses comprehensive characterization of humi-

[Figure]

clike substances (HULIS) in PM2.5 samples from combustions of biomass materials (rice straw, corn straw, and pine branch) and fossil fuels (lignite coal and diesel fuel), and from ambient air. To achieve the goals of this study, water-soluble HULIS fractions were group isolated using a HLB solid phase extraction method and then quantified with a TOC analyzer. Also chemical properties and structures of HULIS were further investigated using elemental analyzer, UV-vis spectroscopy, excitation-emissions matrix (EEM) fluorescence spectroscopy, FTIR spectroscopy, and 1H-NMR spectroscopy. Characteristics of primary HULIS from biomass burning (BB) and fossil fuel (FF) combustion emissions were compared with the result from ambient samples and with those reported from many previous publications. Results indicate the chemical properties and structures of primary HULIS from combustion emissions of biomass and fossil fuels are very similar to chemical features of ambient HULIS in this and previous studies, which are indicated by a variety of analytical tools, with some distinct differences. It is worthy of note that primary HULIS contain mostly low molecular weight compounds. Results from this study can add to the database of chemical properties and structures for BB and FF-derived HULIS, and thereby contribute to better understanding of the role of BB and FF aerosols in ambient environments. Also this work may help to identify future focus in related to molecular level characterization of ambient brown carbon. However, most of the findings from this study were demonstrated by previous publications. Thus, authors should address the unique scientific finding of this work a bit more in revised manuscript. Overall the manuscript is written well, and with some further explanation of collected data and further elaboration on the results it will be ready for publication. Below are specific revision comments for the authors to consider in their next revision.

Reply: Thanks for the comments. We have carefully revised that and address the unique scientific finding of this work a bit more in revised manuscript. The detailed explanation could be found in our reply to referee #1, #2, and the revised manuscript.

Specific comments:

Abstract section I would suggest adding important quantitative information from the study.

Reply: Thanks. We have added some important quantitative information in revised manuscript. The details are as follows:

Page 1, lines 17-20: "The results show that HULIS account for 11.2–23.4% and 5.3% of PM2.5 emitted from biomass burning (BB) and coal combustion, respectively. In addition, contributions of HULIS-C to total carbon and water soluble carbon in smoke PM2.5 emitted from BB and coal combustion are 8.0–21.7% and 5.2%, 56.9–66.1% and 45.5%, respectively."

Page 2, lines 8-15: "HULIS from coal combustion had a lower O/C molar ratio (0.27), and a higher content of [Ar–H] (31%), suggesting that aromatic compounds were extremely abundant in HULIS from this source. Moreover, the absorption Ångström exponents of primary HULIS from BB and coal combustion were 6.7–8.2 and 13.6, respectively. The mass absorption efficiencies of primary HULIS from BB and coal combustion at 365 nm (MAE365) were 0.97–2.09 and 0.63 m2/gC, respectively. Noticeable higher MAE365 for primary HULIS from BB than coal combustion indicate the former one has stronger contribution to the light absorbing properties of aerosols in atmospheric environment."

1. Introduction section Park and Yu (2016) examined the chemical and light absorption properties of HULIS in PM2.5 from burning of three different types of biomass burning fuels (rice straw, pine needles, and sesame branch) in a laboratory combustion chamber ("Chemical and light absorption properties of humic-like substances from biomass burning emissions under controlled combustion experiments". Atmospheric Environment 136, 114-122). Authors may refer to the paper to compare their results.

Reply: Thanks. This is an excellent paper. We have added this reference and make some comparisons in revised manuscript. The details are as follow:

Page 4, lines 3-12: "On recently study, the chemical and light absorption properties of HULIS in PM2.5 from burning of three different types of biomass burning fuels (rice straw, pine needles, and sesame branch) in a laboratory combustion chamber were investigated by Park and Yu (2016). According to this study, primary HULIS from BB accounted for 15.3–29.5% of PM2.5 emissions, and HULIS-C contributed 15–29% of OC and 36–63% of WSOC, respectively. Although the study brought a better understanding on light absorption properties of primary WSOC from BB, but the observation on the optical and structural features of primary HULIS is limited (Park and Yu, 2016). On the other hand, as important energy resources, fossil fuels (such as coal, diesel fuel) are consumed significantly around the world, and to be important sources of black carbon in ambient aerosols (Cao et al., 2006). However, the content and chemical properties of primary HULIS from fossil fuels combustion are still unknown. "

Page 11, lines 17-21: "It can be seen that the mass of the HULIS fractions accounted for 11.2–23.4% of the PM in smoke PM2.5 emitted from BB, which is comparable to the results (7.6–29.5%) for BB reported in previous studies (Lin, 2010b, Park and Yu, 2016). It is worth noting that the highest HULIS abundance (23.4 $\pm$ 5.5%) was detected in rice straw smoke PM2.5, which is consistent with 29.5% for similar samples observed by Park and Yu (2016)."

Page 12, line 16-line 18: "These results are very consistent with the results reported for BB in previous studies (Schmidl et al., 2008a,b; Goncalves et al., 2010; Lin et al., 2010b; Park and Yu, 2016)."

We also have added some valuable data (red) from this reference in revised Table 2.

The added reference is:

Park, S. S., and Yu, J.: Chemical and light absorption properties of humic-like substances from biomass burning emissions under controlled combustion experiments, Atmos. Environ., 136, 114-122, 10.1016/j.atmosenv.2016.04.022, 2016.

2. Experimental section 2.1. Sampling (pages 4-5) Lines 11-12 on page 4: It is described that "..five types of smoke PM2.5 samples were collected to . . .from the combustion of biomass .." How many sets of PM2.5 samples did you collect for each of biomass types and coal fuel? Need to be added in the revision. Was only one sample used for each burn to investigate the comprehensive characterization of HULIS in smoke aerosols samples? If so, they should describe the reliability and/or uncertainties of the experimental results. Also how many ambient samples did you use to conduct the experiments?

Reply: Thanks for comments. We have added some descriptions on sampling of smoke PM2.5 and ambient PM2.5 in revised manuscript. In the study, five sets of PM2.5 samples were collected for each of biomass types and coal fuel. Then five filters in different sets were chosen from different sets for each type of smoke PM2.5 and ambient PM2.5 and were used to investigate the comprehensive characterization of HULIS. The details of sample number have been also added in the revised Table 2. The sentences have been revised as follow:

Page 6, lines 8-17: "The combustion experiments of biomass fuels (rice straw, corn straw and pine branch) were carried out under open air without any controlled conditions to simulate open burning in the field. Smoke PM2.5 samples were collected on Whatman quartz filters (Ø 90 mm) by two samplers in the chamber. For each biomass combustion experiment, biomass materials were firstly cut into pieces, and then were ignited and burned out, and one set of two smoke PM2.5 filters were collected during whole burning process (5∼15 min). Totally, five sets filter samples were collected for each biomass fuel. The coal combustion was carried out according to the method introduced by Huang et al. (2013). The combustion stove was put into the chamber when the combustion condition was stabled, and then one set of smoke PM2.5 sample was collected for approximately 10 min, and a total of five sets of filter samples were obtained."

Page 6, lines 21-23: "In addition to the smoke PM2.5 samples emitted directly from the

combustion process, ambient PM2.5 samples were also collected during December 7 to 11, 2015 in Wushan, Guangzhou, China. Each sample was collected for approximately 24h, and a total of 5 filters were obtained."

Combustion conditions of three biomass materials and lignite coal in a laboratory re-suspension chamber should be described in detail because the burning conditions such as smoldering or flaming burns, combustion temperature, air dilution ratio, flue gas temperature at a sampling location, etc., affect greatly the abundance and chemical properties and structures of WSOC, HULIS, and organic compounds. Also burning conditions might generate water-soluble aerosols of different optical properties. Details in this regard would be helpful. At what stage of the burning were the samples collected? Please be as specific as possible.

Reply: Thanks for comments. This is a good idea, however the smoke PM2.5 samples were collected from uncontrolled combustion in current study. We believed that the studies of HULIS formed from different burning conditions such as smoldering or flaming burns, combustion temperature, air dilution ratio, flue gas temperature at a sampling location, etc. would be very interesting works. Thanks for the advices.

In the current manuscript, we added some descriptions on sampling in the experimental section:

Page 6, line 8-line 17: "The combustion experiments of biomass fuels (rice straw, corn straw and pine branch) were carried out under open air without any controlled conditions to simulate open burning in the field. Smoke PM2.5 samples were collected on Whatman quartz filters (Ø 90 mm) by two samplers in the chamber. For each biomass combustion experiment, biomass materials were firstly cut into pieces, and then were ignited and burned out, and one set of two smoke PM2.5 filters were collected during whole burning process (5∼15 min). Totally, five sets filter samples were collected for each biomass fuel. The coal combustion was carried out according to the method introduced by Huang et al. (2013). The combustion stove was put into the chamber

when the combustion condition was stabled, and then one set of smoke PM2.5 sample was collected for approximately 10 min, and a total of five sets of filter samples were obtained."

What were the moisture contents of the biomass burning and coal fuels? The authors need to describe the elemental composition (C, H, O, N, and S) of burning fuels if possible, but for comparison with other papers moisture content would be very helpful.

Reply: Thanks. We have revised the manuscript according to the comments and added the moisture contents, elemental composition (C, H, O, N, and S) of burning fuels in revised manuscripts.

Page 5, line 20- Page 6, line 4: "The biomass materials including rice straw, corn straw, pine branch were collected from rural area of Guangdong province, and the coal (RO = 0.77%) were obtained from Ping Ding Shan, China. The detail information of this type of coal could be found in Huang et al. (2013). The ultimate properties of the three biomass materials and coal are shown in Table 1. On an air-dry basis, moisture content measured for the rice straw, corn straw, pine branch and coal was $5.8 \pm 0.5$, $7.4 \pm 0.8$, $7.6 \pm 0.7$, and $1.6 \pm 0.2$ %, respectively. Carbon (C), hydrogen (H), and oxygen (O) contents were found to range from 36.0 to 72.6%, 4.0 to 7.2%, and 8.2 to 45.0% for combustion materials, respectively. In comparison with biomass materials, coal substantially comprised of higher C content (72.6%) and lower O content (8.2%). There were no significant differences among biomass materials in terms of elemental compositions."

Moreover, the moisture contents, elemental compositions (C, H, O, N, and S) of burning fuels also have been added in a new Table 1 in current manuscript.

Please add collection time for biomass smokes.

Reply: The collection time is not fixed at a constant time. For each BB combustion experiment, the sampling was conducted until it burned out (5-15 min). The combustion

experiment of coal was conducted according to the method introduced by Huang et al. (2013), and one set of smoke PM2.5 samples was collected for approximately 10 min after the combustion condition was stabled. The sentence has been revised as follow:

Page 6, line 11-line 17: "For each biomass combustion experiment, biomass materials were firstly cut into pieces, and then were ignited and burned out, and one set of two smoke PM2.5 filters were collected during whole burning process (5∼15 min). Totally, five sets filter samples were collected for each biomass fuel. The coal combustion was carried out according to the method introduced by Huang et al. (2013). The combustion stove was put into the chamber when the combustion condition was stabled, and then one set of smoke PM2.5 sample was collected for approximately 10 min, and a total of five sets of filter samples were obtained."

2.2. Isolation of HULIS Lines 5-6 on page 6: ". . .more filters were used to obtain HULIS for the analysis of the elemental composition. . ." Instead of using the HULIS samples re-dissolved in 20 mL Milli-Q water (section 2.2), new filter samples were used for further analyses? More detailed description would be helpful for readers.

Reply: No. The "more filters were used to obtain HULIS for the analysis of the elemental composition. . ." are not the new filter samples. They are a part remainder of filters that have been measured for the quantification of HULIS. We have revised that as follow:

Page 7, line 16-line 18: "Moreover, more area of the corresponding filters was used to obtain enough dried HULIS for the analysis of the elemental composition, as well as FTIR and 1H NMR spectrometry."

2.3.2. Elemental composition Lines 20-21 on page 6: "A portion of the HULIS (redissolved in methanol) was transferred into . . .." The HULIS eluate used in this analysis was not re-dissolved in water? How much volume of MeOH did you use for this?

Reply: In the study, the HULIS was evaporated to dryness under a gentle nitrogen

stream. However, the resulting dried HULIS were not fully in the form of solid powder, but viscous substances. This dried HULIS sample can't be re-dissolved in pure water. Therefore, the HULIS samples were re-dissolved in methanol for elemental analysis in the study. The operate procedure are as follow:

In the experiment, the HULIS sample was re-dissolved them in 3 mL of methanol, then one or two droplets (∼1 mg dried HULIS) was transferred into to a pre-cleaned tin capsule of known weight and dried under vacuum. The mass of the dried HULIS in the tin capsule was determined using a micro balance and then the elemental composition was determined by the elemental analyzer.

2.3.6. 1H-NMR spectroscopy "About 10 mg of dried HULIS were re-dissolved in 1 mL of MeOD." could be changed to ". . ..of MeOH."

Reply: We have revised the sentence as follow:

Page 11, lines 6-7: "About 10 mg of dried HULIS were dissolved in deuterated methanol (MeOH-d4, 1 mL) and transferred to 5 mm NMR tubes."

3. Results and discussion 3.1. The abundance of HULIS in smoke PM2.5 and ambient PM2.5 These results should be compared with those from Park and Yu (2016). In Table 1, please include number of samples used in the experiments for each of BB, FF, and ambient samples. References of Park et al. in Table 1 are not listed in the list of the references.

Reply: Thanks. We have revised the manuscript according to the comments. The details are as follow:

Page 11, lines 19-21: "It is worth noting that the highest HULIS abundance (23.4 ± 5.5%) was detected in rice straw smoke PM2.5, which is consistent with 29.5% for similar samples observed by Park and Yu (2016)."

Page 12, lines 16-18: "These results are very consistent with the results reported for BB in previous studies (Schmidl et al., 2008a, b; Goncalves et al., 2010; Lin et al.,

2010b; Park and Yu, 2016)."

In addition, we have added the number of samples used in the experiments for each of BB, FF, and ambient samples in revised Table 2 (i.e., the old Table 1), and also added some valuable data from Park and Yu (2016) in revised Table 2 (i.e., the old Table 1).

Finally, the references of "References of Park et al. in Table 1" have been added in revised manuscript:

Page 33, Lines 22-24: Park, S. S., Cho, S. Y., Kim, K. W., Lee, K. H., and Jung, K.: Investigation of organic aerosol sources using fractionated water-soluble organic carbon measured at an urban site, Atmos. Environ., 55, 64-72, DOI 10.1016/j.atmosenv.2012.03.018, 2012.

Page 33, Lines 25-26: Park, S. S., and Cho, S. Y.: Characterization of Organic Aerosol Particles Observed during Asian Dust Events in Spring 2010, Aerosol Air Qual. Res., 13, 1019-1033, DOI 10.4209/aaqr.2012.06.0142, 2013.

Page 33, Lines 27-28: Park, S. S., Schauer, J. J., and Cho, S. Y.: Sources and their contribution to two water-soluble organic carbon fractions at a roadway site, Atmos. Environ., 77, 348-357, DOI 10.1016/j.atmosenv.2013.05.032, 2013.

Page 33, Lines 29-30: Park, S. S., and Yu, J.: Chemical and light absorption properties of humic-like substances from biomass burning emissions under controlled combustion experiments, Atmos. Environ., 136, 114-122, 10.1016/j.atmosenv.2016.04.022, 2016.

3.2. Elemental composition Lines 15-21 on page 11: In Table 2, OM/OC ratios for four types of primary HULIS are presented. They did not measure OC concentration. Details how OM/OC ratios got determined from elemental composition data should described in the text.

Reply: Thanks. We have added a description for the OM/OC determination in Table 3 of revised manuscript.

In the study, OM/OC represents the organic matter-to-organic carbon mass ratio. The OM is referred to the mass of HULIS, which has been determined with microbalance. The OC is referred to mass of carbon content of HULIS, which has been measured with elemental analyzer. Finally, the OM/OC ratio was calculated from the mass ratio of OM to OC.

3.3 UV-vis properties & 3.4 Fluorescence properties I think that authors measured light absorption spectra of WSOC and HULIS from BB, FF, and ambient samples. I would suggest providing absorption angstrom exponents (AAE) and mass absorption efficiencies (MAE) of samples from burning of different types of biomass and coal fuels, and ambient environment. These information could be much useful for understanding light absorption characteristics and radiative forcing effects by BB and coal burning derived brown carbon aerosols.

Reply: Thanks for the comments. We have added the discussion of AAE and MAE of HULIS samples from burning of different types of biomass and coal fuel, and from ambient aerosols in revised manuscript. We clearly stated in our revised manuscript as below:

Page 2, lines 10-15: "Moreover, the absorption Ångström exponents of primary HULIS from BB and coal combustion were 6.7–8.2 and 13.6, respectively. The mass absorption efficiencies of primary HULIS from BB and coal combustion at 365 nm (MAE365) were 0.97–2.09 and 0.63 m2/gC, respectively. Noticeable higher MAE365 for primary HULIS from BB than coal combustion indicate the former one has stronger contribution to the light absorbing properties of aerosols in atmospheric environment."

Page 9, line 21- Page 10, line 12: a new section of "(3) Light absorption properties . . .. . .." was added in the experimental section of revised manuscript.

Page 24, line 22- Page 26, line 12: A new section of "3.7 Light absorption properties" was added in the results and discussion section in revised manuscript.

Page 27, lines 8-9: "The AAE and MAE365 of the BB derived primary HULIS were similar to those of atmospheric HULIS and/or WSOM."

Page 28, lines 1-3: "The MAE365 of BB HULIS are 0.97-2.09 m2/g, which are higher than that of coal combustion HULIS, suggesting the former one own stronger light absorption properties."

We also added a new Table 7 "Summary of AAE and MAE365 of HULIS and WSOM" in revised manuscript.

Moreover, some new references have been also added in revised manuscript. (Page 29, lines 26-27; Page 29, lines 28-30; Page 29, lines 31-Page 30, line 1; Page 30, lines 26-28; Page 32, lines 5-7; Page 32, lines 8-9; Page 32, lines 15-17; Page 32, lines 18-20; Page 33, lines 1-3; Page 33, lines 29-30; Page 35, lines 12-14; Page 35, lines 15-17)

Lines 19-20 on page 13 and lines 10-11 on page 15. Authors stated that based on the SUVA254 values from primary smoke HULUS, "the primary HULIS contained higher aromatic degree and/or higher MW compounds", but results from EEM spectra indicate that "primary HULIS contain mostly low MW compounds". This means that primary HULIS from BB and FF smokes contain both high and low MW compounds? Further elaboration on this is needed.

Reply: Thanks. In order to avoid the misunderstanding, these sentences are revised as follows:

Page 16, lines 22-24: "These results indicate that the primary HULIS contain more aromatic groups with conjugation of $\pi$-bonds alongside aliphatic structures."

Page 18, lines 17-20: "This finding indicates that these four types of primary HULIS are consist of more phenol-like, protein-like, and/or aromatic amino acids than atmospheric HULIS (Coble, 1996; Peuravuori et al., 2002; Duarte et al., 2004; Kieber et al., 2006)."

3.7 Comparison pf primary HULIS and 4 Conclusions Sections 3.7.1, 3.7.2, and 3.7.3

are very similar to the explanations in sections 3.2-3.6, so it needs to be condensed, or I suggest combining the section 3.7 with section 4. Conclusions.

Reply: Thanks for the comments. This is a good idea. We have revised that in current manuscript. The section 3.7 has been combined with section 4. Conclusions in revised manuscript. The revisions are as follow:

Page 26, line 14-page 28, line 6: "4. Conclusions. . . . . ." in revised manuscript

4. Conclusions It will be much more valuable if a paragraph was added to conclusions describing what the authors think was important and how it can be applied

Reply: We have added a new paragraph to introduce the implication in revised manuscript. The revisions are as follow:

Page 28, lines 8-Page 29, line 1: "5 Implications. . . . . . ." was added in revised manuscript.

---

## Editor Decision (ED1)

Requested revisions to ACP-2016-397

A number of the new sections that you added either have mistaken grammar, or are not clear. So I'm asking you to fix those things before the paper can be accepted.

The pages and line number refer to the corrected manuscript.

Page 1, Lines 18-20, I can't parse this phrase to figure out what the numbers refer to. Please split this up to be explicit.

Page 2, Line 13. 'indicate' should be 'indicates'
Page 2, Line 14. Should be 'the atmospheric environment'.
Page 4, Line 3. 'On recently study', should be 'One recent study'
Page 4, Line 8. 'but the observation on… is limited' should be 'but the observations of… are limited"
Page 4 Lines 10-11. 'and to be important' should be and are known to be important'
Page 5, Line 10. 'branches'
Page 5, Line 21. What does Ro = 0.77% mean?
Page 5, Line 22. 'of' should be 'on'
Page 6, Line 2. 'comprised' should be 'composed'
Page 6, Line 12. 'firstly' should be 'first'
Page 6, Line 14. 'sets filter' should be 'sets of filters'
Page 6, Line 16. 'stabled' should be 'stabilized'
Page 7, Line 22. 'differences were observed' should be 'differences that were observed'
Page 9, Line 18. 'be' should be 'been'
Page 14, Line 16. 'dropped' should be 'fell'
Page 15, Line 5. Eliminate the word 'of'
Page 16, Line 24. 'characters' should be 'characteristics'
Page 18, Line 18. Eliminate the word 'are'
Page 20, Line 6. Instead of 'present less' consider using 'contain fewer'
Page 21, Line 17. 'deeply' should be 'deep'
Page 22, Lines 19-20. The first part of the sentence doesn't make sense. Are you trying to say that both BB and Coal-derived HULIS exhibit many sharp signals between 6.5-8.5 ppm?
Page 23, Lines 23-24. I can't tell what you mean by this sentence.
Page 27, Line 12. Instead of 'were characterized as contain' it would be better to say 'contained'
Page 28, Lines 15-16. It is not clear what you mean when you say 'was firstly' or 'were firstly'. Are you saying this is the first time these features have been observed? If so, you will need to say it differently.

---

## Author Response (AR2)

***Reply to comments on*** **"Comprehensive characterization of humic-like substances in smoke PM$_{2.5}$ emitted from the combustion of biomass materials and fossil fuels"** ***by*** **Xingjun Fan et al.**

Dear Dr. James Roberts,

We are grateful to your valuable comments, and we have carefully revised our manuscript accordingly. The point-to-point responses to the comments are given below.

Requested revisions to ACP-2016-397

A number of the new sections that you added either have mistaken grammar, or are not clear. So I'm asking you to fix those things before the paper can be accepted.

The pages and line number refer to the corrected manuscript.

Page 1, Lines 18-20, I can't parse this phrase to figure out what the numbers refer to. Please split this up to be explicit.

Reply: Thanks for the comments. We have revised the sentence as follow:

Page 1, Lines 18-21: "In addition, contributions of HULIS-C to total carbon and water soluble carbon in smoke PM$_{2.5}$ emitted from BB are 8.0–21.7% and 56.9–66.1%, respectively. The corresponding contributions in smoke PM$_{2.5}$ from coal combustion are 5.2% and 45.5%, respectively."

Page 2, Line 13. 'indicate' should be 'indicates'

Reply: Revised. (Page 2, Line 14)

Page 2, Line 14. Should be 'the atmospheric environment'.

Reply: Revised. (Page 2, Line 15)

Page 4, Line 3. 'On recently study', should be 'One recent study'

Reply: Revised. (Page 4, Line 3)

Page 4, Line 8. 'but the observation on… is limited' should be 'but the observations of… are limited"

Reply: Revised. (Page 4, Line 8-9)

Page 4 Lines 10-11. 'and to be important' should be and are known to be important'

Reply: Revised. (Page 4, Line 10-11)

Page 5, Line 10. 'branches'

Reply: Revised. (Page 5, Line 10)

Page 5, Line 21. What does Ro = 0.77% mean?

Reply: In the study, the $R_o(\%)$ is the vitrinite reflectance of the coal, which indicate the thermal maturity or rank of coals. In order to avoid misunderstanding, the

abbreviation ($R_o$) has been revised as follow:

Page 5, Lines 20-22. "The biomass materials including rice straw, corn straw, pine branches were collected from rural area of Guangdong province, and the coal (the vitrinite reflectance ($R_o$) is 0.77%) were obtained from Ping Ding Shan, China."

Page 5, Line 22. 'of' should be 'on'

Reply: Revised. (Page 5, Line 22)

Page 6, Line 2. 'comprised' should be 'composed'

Reply: Revised. (Page 6, Line 2)

Page 6, Line 12. 'firstly' should be 'first'

Reply: Revised. (Page 6, Line 12)

Page 6, Line 14. 'sets filter' should be 'sets of filters'

Reply: Revised. (Page 6, Line 14)

Page 6, Line 16. 'stabled' should be 'stabilized'

Reply: Revised. (Page 6, Line 16)

Page 7, Line 22. 'differences were observed' should be 'differences that were observed'

Reply: Revised. (Page 7, Line 22)

Page 9, Line 18. 'be' should be 'been'

Reply: Revised. (Page 9, Line 18)

Page 14, Line 16. 'dropped' should be 'fell'

Reply: Revised. (Page 14, Line 16)

Page 15, Line 5. Eliminate the word 'of'

Reply: Revised.

Page 16, Line 24. 'characters' should be 'characteristics'

Reply: Revised. (Page 16, Line 24)

Page 18, Line 18. Eliminate the word 'are'

Reply: Revised.

Page 20, Line 6. Instead of 'present less' consider using 'contain fewer'

Reply: Revised. (Page 20, Line 6)

Page 21, Line 17. 'deeply' should be 'deep'

Reply: Revised. (Page 21, Line 17)

Page 22, Lines 19-20. The first part of the sentence doesn't make sense. Are you trying to say that both BB and Coal-derived HULIS exhibit many sharp signals

between 6.5-8.5 ppm?

Reply: Thanks for the comment. Yes, we want to say that both BB and Coal-derived HULIS exhibit many sharp signals between 6.5-8.5 ppm. In the current manuscript, the sentence has been revised as follow:

Page 22, lines 19-22: "Moreover, both BB and coal combustion derived HULIS exhibit many sharp signals between 6.5–8.5 ppm, which could ascribed to aromatic structures, such as substituted phenols and alkylbenzenes (around 6.6–7.0 ppm), benzoic acids, esters, and nitroaromatics (Suzuki et al., 2001; Chalbot et al., 2014)."

Page 23, Lines 23-24. I can't tell what you mean by this sentence.

Reply: Thanks for the comments. We have revised the sentence as follow:

Page 23, Lines 20-24: "The atmospheric HULIS in ambient aerosols from this work and other studies (Song et al., 2012; Fan et al., 2013; Lopes et al., 2015) and in rainwater (Miller et al., 2009; Santos et al., 2009, 2012) were all characterized by the highest content of [H–C] (37–60%), moderate contents of [H–C–C=] (20–37%) and [H–C–O] (10–24%), and the lowest content of [Ar–H] (1–12%)."

Page 27, Line 12. Instead of 'were characterized as contain' it would be better to say 'contained'

Reply: Revised. (Page 27, Line 12)

Page 28, Lines 15-16. It is not clear what you mean when you say 'was firstly' or 'were firstly'. Are you saying this is the first time these features have been observed? If so, you will need to say it differently.

Reply: Thanks for the comments. In the study, we want to say that coal combustion was firstly identified as an important source of primary HULIS. In order to avoid the misunderstanding, the sentences have been revised as follow:

[revised manuscript text omitted]

[a] Representing the range of wavelength chosen for fitting.